# A BRET biosensor for measuring uncompetitive engagement of PRMT5 complexes in cells

Elisabeth M. Rothweiler[1,2,9], Ani Michaud[3,9], Jakub Stefaniak[1,2], Usha Singh[1,2], Brynwood B. Mikulsky[3], James D. Vasta[3], Michael T. Beck[3], Jennifer Wilkinson [3], Jennifer A. Ward[1,2], Catherine M. Rogers[1,2], Esra Balıkçı [1,2], Jeppe Tranberg-Jensen[1,2], Jesper S. Hansen[1,2], Peter Loppnau[4], Adrian Whitty [5], Paul E. Brennan[1,2,6], Peter J. Tonge [7,8] ✉, Matthew B. Robers [3] ✉ & Kilian V. M. Huber [1,2] ✉

Protein arginine methyl transferase 5 (PRMT5) plays a global role in cell physiology and is an established therapeutic target in cancer. In approximately 10-15% of human cancers, deletion of the methylthioadenosine phosphorylase (MTAP) gene results in accumulation of methylthioadenosine (MTA), exposing a synthetic lethality and opportunity for precision medicine by selective targeting of PRMT5 in this context. Reported small molecule PRMT5 inhibitors engage either cosubstrate S-adenosyl methionine (SAM) or peptide-substrate pockets through diverse mechanisms. A subset of chemotypes demonstrate uncompetitive engagement with SAM or its inhibitory metabolic precursor, MTA. Although uncompetitive engagement can be evaluated in cell-free systems, no methods exist to directly assess this in cells. Here, we describe the development of a fluorescent probe that acts as a dynamic BRET biosensor of the intracellular SAM/MTA pool that overcomes the current limitations of competitive binding analyses. Using this biosensor, we evaluate a range of diverse PRMT5 inhibitors to mechanistically characterize and quantify uncompetitive target engagement as well as ternary complex formation at PRMT5-SAM and PRMT5-MTA complexes in live cells, enabling direct insights into drug mechanism-of-action and metabolite-dependent responses of inhibitors.

Protein arginine methyl transferases (PRMTs) transfer a methyl group from their cosubstrate S-adenosyl-methionine (SAM) to peptide substrates. There are nine mammalian PRMTs that are classified by their mode of methyl group transfer into three sub-types. Type II PRMTs, PRMT5 and PRMT9, mediate symmetrical dimethylation of arginine through addition of a methyl group to the ω-nitrogen of an arginine residue[1]. PRMT5 regulates a number of cellular processes, including transcription, nucleosome remodeling and the formation of corepressor complexes[2]. PRMT5 distinguishes its methylation targets by interacting with nuclear and cytosolic adapter proteins such as WDR77

[1]Centre for Medicines Discovery, Nuffield Department of Medicine, University of Oxford, Old Road Campus, Oxford, UK. [2]Target Discovery Institute, Nuffield Department of Medicine, University of Oxford, Old Road Campus, Oxford, UK. [3]Promega Corporation, Madison, WI, USA. [4]Structural Genomics Consortium, University of Toronto, Toronto, ON, Canada. [5]Department of Chemistry, Boston University, Boston, MA, USA. [6]Alzheimer's Research UK Oxford Drug Discovery Institute, Nuffield Department of Medicine, University of Oxford, Oxford, UK. [7]Center for Advanced Study of Drug Action, Department of Chemistry, Stony Brook University, Stony Brook, NY, USA. [8]Department of Biomedical Genetics, University of Rochester, Rochester, NY, USA. [9]These authors contributed equally: Elisabeth M. Rothweiler, Ani Michaud. ✉e-mail: peter.tonge@stonybrook.edu; matt.robers@promega.com; kilian.huber@cmd.ox.ac.uk

(MEP50)[3]. There is substantial evidence that PRMT5 plays a pivotal role in the pathogenesis of hematologic and solid tumor malignancies[4]. The PRMT5 peptide-substrate and cosubstrate (SAM)-binding pockets, which are connected through a central channel, are vulnerable to small molecule inhibitors[5,6], and PRMT5 inhibitors can be classified based on their pocket specificity and mode of action. First-generation PRMT5 inhibitors engage either the cosubstrate or peptide-substrate pockets through mechanisms that are either competitive or uncompetitive with SAM. The adenosine analogs LLY-283 and PF-06939999 represent some of the first described SAM-competitive inhibitors[7–10]. The peptide-substrate binding pocket is structurally diverse as PRMTs methylate a range of proteins, affording distinct opportunities for targeting PRMT5. EPZ015666 and GSK3326595 (pemrametostat), are highly potent PRMT5 inhibitors that uniquely engage the PRMT5-SAM complex[11]. Unlike the mutually exclusive mode of action for cosubstrate pocket inhibitors, structural and enzymatic studies support a

peptide-substrate-competitive, SAM-uncompetitive mode of action for EPZ015666 and GSK3326595[11,12]. Several substrate-competitive inhibitors have entered clinical trials[13,14], however, these first generation PRMT5 inhibitors that unconditionally target the cosubstrate and peptide-substrate pockets are constrained by their limited therapeutic window. More recently, new precision medicine opportunities have emerged through next-generation PRMT5 inhibitors that exploit the synthetic-lethal relationship between PRMT5 and methylthioadenosine phosphorylase (MTAP). PRMT5 activity is tightly regulated by the intracellular concentration of SAM, and intracellular SAM homeostasis is maintained through key enzymes that salvage methionine from other cellular reactions for SAM biosynthesis (Fig. 1A). Methionine adenosyltransferase 2A (MAT2A) catalyzes the rate-limiting step in the biosynthesis of SAM from ATP and methionine. To maintain methionine levels, the methionine salvage pathway rescues a key metabolic intermediate, methylthioadenosine (MTA) from adjacent pathways to

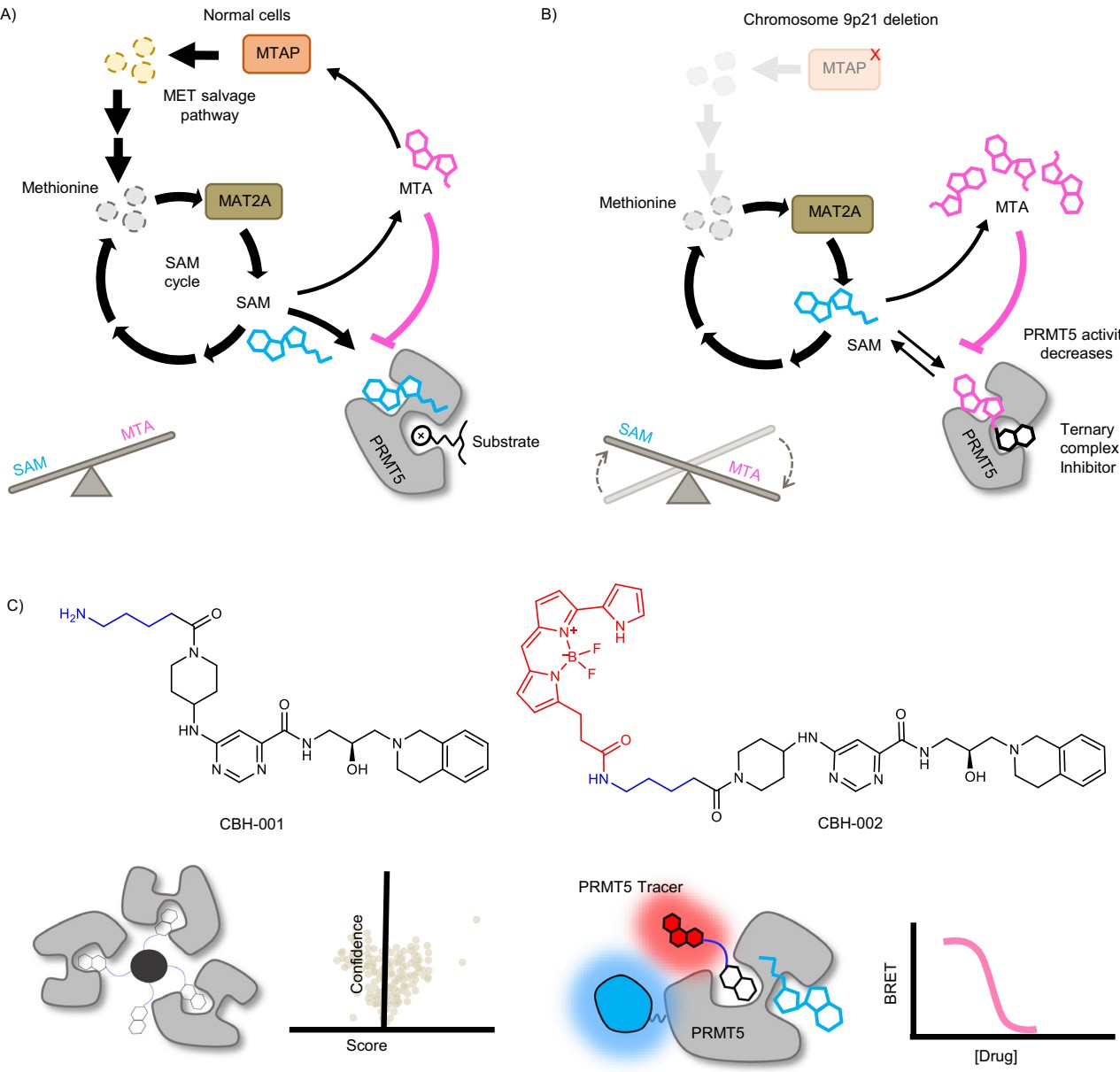

**Fig. 1 | Intracellular regulation of PRMT5 activity and development of chemical probes CBH-001 and CBH-002. A** Wild-type cells have high levels of SAM and relatively low levels of MTA, due to concurrent SAM-biosynthesis (driven by MAT2A) and methionine salvage (driven by MTAP); here, PRMT5 is predominantly SAM-bound and active. **B** MTAP deletion leads to an accumulation of intracellular

MTA, which eventually out-competes SAM for binding to PRMT5, reducing methyltransferase activity. MTA-cooperative inhibitors have high affinity for MTA-bound PRMT5, causing selective cancer cell killing. **C** Structures of the chemoproteomic affinity probe CBH-001 and BRET biosensor CBH-002 for NanoBRET target engagement assays.

generate methionine for SAM biosynthesis. MTAP mediates an essential step by phosphorylating MTA to yield 5′-methylthioribose-1-phosphate, which can be recycled into methionine to maintain SAM levels[15,16]. In 10-15% of cancers, homozygous deletion of MTAP results in accumulation of MTA, which competes with SAM for the PRMT5 cofactor binding pocket, inhibiting methyltransferase activity (Fig. 1B)[17]. Accordingly, the loss of MTAP in cancer cells exposes several collateral vulnerabilities in the SAM-PRMT5 pathway, including MAT2A and PRMT5[18,19]. Inhibitors that uncompetitively engage the PRMT5-MTA complex can exploit this synthetic lethality and offer selectivity in MTAP-deleted tumors, presenting an intriguing opportunity for personalized genomic medicine[16,20–22]. MRTX1719 was characterized as a substrate-competitive, MTA-cooperative inhibitor with selectivity for MTAP-deleted cancer cells[21]. Additional molecules have been reported that inhibit PRMT5 by forming a ternary complex with PRMT5-MTA, which are at various stages of preclinical and clinical development[22–26]. Although being broadly categorized as MTA-cooperative, where cooperative is taken to describe that MTA enhances the affinity of the inhibitors for the enzyme, these inhibitors demonstrate varying sensitivity to genetic ablation of the MTAP gene. These observations support the possibility that PRMT5-MTA ternary complex inhibitors may have varying degrees of MTA-sensitivity in cells, or that the intracellular concentrations of cosubstrates or other cellular factors likely underlie PRMT5 vulnerability. Despite pronounced selectivity in MTAP-deleted cancer cells, the molecular mode of action for MTA-cooperative inhibitors cannot be directly measured in a pathophysiological setting. To determine whether such ternary complex inhibitors engage PRMT5 through an uncompetitive mechanism in cells, methods are required for surveying uncompetitive target engagement within live cells. This represents a critical gap and current limitation for chemical biology and synthetic lethal drug discovery. BRET probes have enabled target occupancy analysis for various protein targets in live cells[27–30], and to date, are applied in displacement assay formats to query competitive target engagement, such as broad-spectrum kinase profiling. However, a significant limitation has been the direct assessment of more complex inhibition mechanisms like uncompetitive engagement in live cells.

Here, we describe a cell-permeable PRMT5 BRET probe, CBH-002, that is highly sensitive to the intracellular SAM/MTA pool and capable of detecting uncompetitive target engagement at PRMT5 complexes. Using this probe, we present a method to quantify live-cell target engagement for both peptide-substrate and cosubstrate-competitive inhibitors. Notably, the BRET probe allows for direct quantification of MTA- and SAM-uncompetitive PRMT5 engagement, thereby providing direct insight into the mechanism of target inhibition and the relative abundance of SAM and MTA. This quantitative capability can be leveraged to stratify cofactor-sensitivity and intracellular ternary complex formation in MTAP-deleted colorectal cancer cells.

## Results

### Development of a cosubstrate-sensitive PRMT5 BRET probe

Bioluminescence Resonance Energy Transfer (BRET) methods have been successfully used to quantify orthosteric and allosteric target engagement across a variety of intracellular proteins[27,31]. However, to quantify cosubstrate-dependent engagement of ligands that are uncompetitive with respect to MTA or SAM, a PRMT5 BRET probe must demonstrate sensitivity to occupancy of both peptide-substrate and cosubstrate pockets. As a SAM-uncompetitive, peptide-substrate-competitive PRMT5 inhibitor, GSK3326595[12] presents an intriguing scaffold for BRET probe development[12]. Based on previously published PRMT5 inhibitor co-crystal structures[11], we designed and synthesized an amine-functionalized analog of GSK3326595, CBH-001, which we immobilized on sepharose beads to generate a chemoproteomic affinity matrix[32], enabling enrichment of endogenous PRMT5 protein from cells (Fig. 1C). Western blot analysis confirmed potent enrichment of

PRMT5 and its complex partner WDR77 (MEP50) which were both efficiently competed by free unmodified GSK3326595 (Fig. S1A). Subsequent mass spectrometry analysis suggested GSK3326595 to be highly specific for its cognate target with exquisite selectivity observed for PRMT5 alongside known interactors (Fig. S1A). Further characterization by 2D thermal profiling corroborated these results (Fig. S1B) prompting us to synthesize the BRET-compatible probe CBH-002 by appending a BRET acceptor fluorophore (NanoBRET 590SE) to CBH-001 (Fig. 1C).

Next, we evaluated CBH-002 in live HEK293 cells expressing N- or C-terminal fusions of PRMT5 with NanoLuc (NLuc) (Fig. S2A, B). The N-terminal fusion gave a robust and specific BRET signal that could be competed with unmodified GSK3326595, however the dose response curves for both NLuc orientations were biphasic. Recognizing that WDR77 (MEP50) may influence PRMT5 target engagement, and that WDR77 may not be expressed at levels commensurate to that of the PRMT5 expression system, we co-expressed a 10-fold excess of untagged WDR77 DNA together with NLuc-PRMT5. Co-expression of untagged WDR77 potentiated the BRET signal by orders of magnitude compared to NLuc-PRMT5 alone, supporting formation of a more competent methyltransferase complex (Fig. 2A). Furthermore, co-expression of WDR77 mitigated the biphasic behavior of the BRET response yielding a single binding isotherm (Fig. 2A). These results suggest that the higher-order methyltransferase complex is favored when WDR77 is introduced to PRMT5 at saturating levels. CBH-002 did not show engagement at PRMT7, PRMT9, or non-methyltransferase proteins like RIPK2 and BDR9, suggesting good selectivity for PRMT5 (Fig. S2C–F). Correct subcellular localization of NLuc-PRMT5 protein was confirmed via fluorescence imaging (Fig. S2G). Titrations of GSK3326595 were performed at different concentrations of CBH-002, and $EC_{50}$ values were determined by fitting the data to a three-parameter concentration response equation in which Hill slope was constrained to 1 (Fig. 2B). An optimal BRET probe concentration of 30 nM was chosen as it provided adequate signal and was near the BRET probe $EC_{50}$ of 33 nM.

### CBH-002 is sensitive to all modes of PRMT5 target engagement

To explore the broader utility of our BRET system, we evaluated a subset of PRMT5 inhibitors in competitive displacement assays with CBH-002. Remarkably, CBH-002 was able to query all currently described modes of PRMT5 engagement (Fig. 2C–E). NanoBRET binding potencies for GSK3326595 and EPZ015666 were 20 nM and 27 nM, respectively, largely agreeing with both western blot and cellular proliferation data (Fig. 2C, Table S1). LLY-283 and PF-06939999 potencies were 467 nM and 1.7 nM, respectively, and also agreed with published cellular assay data (Fig. 2D, Table S1). The sensitivity of the substrate-competitive BRET probe to cosubstrate-competitive inhibitors such as LLY-283 supports the notion of allosteric communication between these binding sites. Distinct type I and III PRMT family inhibitors did not demonstrate target engagement at PRMT5, supporting the specificity of the PRMT5 BRET method (Fig. S2H).

CBH-002 was also sensitive to engagement of inhibitors such as MRTX1719 and TNG908 that have been designated as MTA-cooperative (Fig. 2E). Published data for MTA-cooperative inhibitors show a disconnect between biochemical and cellular assay readouts, with reported biochemical potencies generally in the double-digit nanomolar range and cellular potencies in MTAP wild-type lines in the micromolar range (Table S1). Interestingly, BRET engagement potencies observed in HEK293 cells for MRTX1719 and TNG908 were more potent than expected for a MTAP wild-type cell-line (80 nM and 135 nM, respectively), indicating that intracellular MTA levels in HEK293 cells were sufficient to support compound binding (Table S1). Weak engagement of such molecules is expected in cell lines with wild-type MTAP, however, cellular potency varies widely, depending on the cell line used[20,22,23,26], suggesting that PRMT5 inhibitor engagement

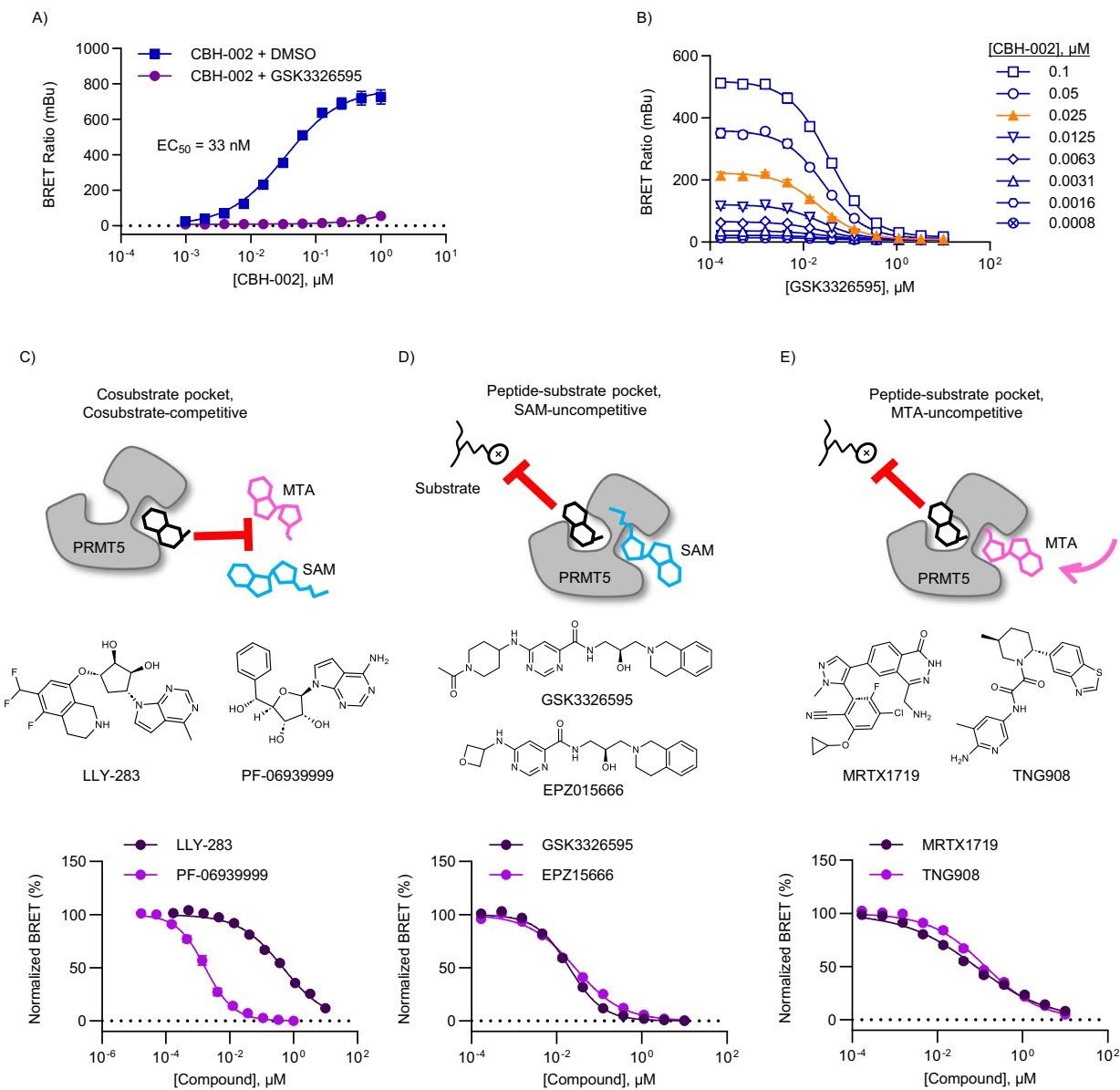

**Fig. 2 | BRET probe CBH-002 enables live-cell target engagement assay for all three subclasses of PRMT5 inhibitors. A** Affinity curve for BRET probe CBH-002 in HEK293 cells transfected with NanoLuc-PRMT5 and unlabeled WDR77 DNA. CBH-002 was titrated against 10 μM GSK3326595 or DMSO, and apparent affinity (EC$_{50}$) was calculated from a sigmoidal 4-parameter curve-fit (see "Methods"). Data are the mean ± SEM of 3 independent experiments (n = 3). **B** Optimization of BRET probe concentration for live-cell TE. An 8 pt, 2-fold series of CBH-002 and an 11 pt, 3-fold dilution series of GSK3326595 were added to HEK293 cells expressing NanoLuc-PRMT5 fusion DNA and unlabeled WDR77 DNA. Optimal BRET probe concentration shown in orange. Data are the mean ± SEM of 3 independent experiments (n = 3). **C–E** Schematic representations of the three major PRMT5 inhibitor subclasses (top) with structures of representative compounds (middle), and live-cell NanoBRET data (bottom). Data are the mean ± SEM of 3 independent experiments (n = 3). Source data are provided as a Source Data file.

potency may be highly context-dependent and impacted by a variety of factors present in the cellular milieu. For example, intracellular engagement may be influenced by the relative SAM/MTA pool, participation of PRMT5 in complexes with other proteins, and/or efflux. To fully characterize the impact of MTA on PRMT5 inhibitor engagement and accurately assess mechanism of action for target engagement, we decided to interrogate PRMT5 inhibitor pharmacology over a wide range of SAM:MTA ratios and physiologically relevant cell models.

### CBH-002 sensitivity to intracellular SAM/MTA levels

Based on the reported mechanism of action for SAM- and MTA-cooperative PRMT5 inhibitors, modulation of the intracellular SAM/MTA pool should directly impact target engagement pharmacology

for all reported PRMT5 inhibitors. We rationalized that CBH-002 should report on intracellular SAM levels, and that it should be competitive with intracellular MTA. Owing to positive charge, SAM is an impermeable compound, therefore a direct assessment of SAM levels on the PRMT5 BRET assay is not possible via titration of exogenous cosubstrate into cells[33]. Therefore, to establish that the BRET probe was directly sensitive to SAM levels, uncompetitive binding of CBH-002 with SAM was evaluated in cell lysates. Indeed, we found the BRET biosensor was highly sensitive to [SAM] over approximately two orders of magnitude (Fig. S3A, B). Based on this linear range of SAM sensitivity, CBH-002 BRET potency may therefore be used to estimate the intracellular levels of SAM in intact cells. Live-cell CBH-002 potency (33 nM, Fig. 2A) fell within the linear dynamic range of SAM titration

experiments, supporting the notion that intracellular [SAM] levels are in the low micromolar range in HEK293 cells, consistent with literature estimates from other cell systems[34]. To further evaluate the intracellular sensitivity of CBH-002 to SAM levels in live cells, we monitored BRET signal following modulation of the SAM biosynthetic pathway via MAT2A inhibition (Fig. 3A, C, D). A series of MAT2A inhibitors, including "Compound A", were recently disclosed in the patent literature (Fig. 3C)[35,36], offering a chemical probe strategy for SAM modulation.

To first understand BRET probe binding kinetics, we evaluated real-time kinetics of CBH-002 association and dissociation in live cells, indicating that the PRMT5 BRET biosensor reaches equilibrium in ~2 h at room temperature (Fig. S3C), suggesting protracted association kinetics. Following pre-equilibration with CBH-002, only a fraction of the BRET signal could be outcompeted with free unmodified GSK3326595, indicating the probe also had slow dissociation kinetics (Fig. S3C). Although this kinetic resolution may not be sufficient to quantify engagement of rapidly equilibrating drugs, we hypothesized it may be useful to evaluate the protracted metabolic response to MAT2A inhibition. Additionally, unlike direct titration of SAM into cell lysates, inhibition of SAM levels via MAT2A inhibition should require active cell metabolism, and thus we performed the BRET assay over a 6 h interval in real time at 37 °C. Using pre-equilibrated CBH-002, we observed dose- and time-dependent BRET inhibition via Compound A treatment in live cells, reaching a plateau at ~5 h at 37 °C (Fig. 3D). The effect of MAT2A inhibition was not evident at room temperature, presumably owing to attenuated SAM metabolism at colder temperatures (Fig. S3D). Together, the SAM titration and MAT2A inhibition results support the sensitivity of CBH-002 to intracellular SAM levels, and corroborate the proposed mechanism of action of MAT2A inhibition. Unlike SAM which potentiates CBH-002 binding, we speculated that MTA would be functionally competitive with BRET probe binding. In contrast to SAM, MTA is highly permeable and is able to enter intact cells[33]. Therefore, to directly assess the impact of MTA levels on the BRET probe signal, we titrated exogenous MTA into the PRMT5 BRET assay in HEK293 cells (Fig. 3B, E). Dose- and time-dependent inhibition of BRET was observed with exogenous MTA, confirming that CBH-002 can detect allosteric binding of MTA in the cosubstrate pocket (Fig. 3E). The effect of MTA was immediately evident as was the dose-dependent inhibition of the BRET signal which saturated rapidly at room temperature (plateauing at ~2 h), supporting the notion of direct MTA competition with the BRET probe. This result is in contrast with MAT2A inhibition via Compound A treatment, that was only evident at 37 °C and did not impact CBH-002 binding at room temperature over a 2-h interval (Fig. S3D), demonstrating the PRMT5 BRET biosensor is capable of resolving direct kinetic mechanisms of PRMT5 engagement versus indirect mechanisms involving SAM metabolism. Together, these results suggest that CBH-002 is a robust biosensor for the intracellular SAM/MTA pool, and that the BRET system should be competent to evaluate target engagement under a range of intracellular MTA and SAM concentrations.

In order to provide a rationale for our findings, we examined a variety of PRMT5 co-crystal structures in complex with inhibitors and different metabolites (Fig. S4). In case of the substrate mimetic EPZ015666, the tetrahydroisoquinoline scaffold plays a crucial role in binding to PRMT5 (PDB: 4×61) (Fig. S4A). The bicyclic ring structure engages with F327 through π-π stacking and forms a cation-π interaction with the positively charged sulfonium group of SAM[11]. In the absence of the inhibitor, SAM predominantly interacts with PRMT5 mainly via the adenosine core (PDB: 7L1G), while the remaining portion of the cofactor extends towards the substrate binding pocket. However, upon binding of EPZ015666, SAM adopts a different conformation to accommodate the inhibitor (Fig. S4B). Notably, F327 adopts distinct positions depending on whether SAM, EPZ015666, LLY-283 or MTA (PDB: 6CKC) is bound (Fig. S4C, D). In case of the MRTX1719:MTA

complex (PDB: 7S1S)[37], a shift in F327 is observed to accommodate MRTX1719 binding (Fig. S4D). This shift, similar to that seen with LLY-283, likely accounts for our finding that CBH-002 is able to quantify TE for MTA-uncompetitive inhibitors.

## Measuring MTA-uncompetitive PRMT5 target engagement in cells

Structural and biochemical evidence supports positive MTA cooperativity and ternary complex formation for MRTX1719 and related chemotypes[21,22]. However, the molecular mode of MTA cooperativity and the impact of the intracellular MTA pool on PRMT5 occupancy has not been evaluated in live cells. Furthermore, although MTA levels are elevated in MTAP-deleted cancers, the different levels of SAM, MTA, and other metabolic intermediates may vary across cells and tissues so that PRMT5 vulnerability may be impacted non-uniformly even in genetically defined backgrounds. Using a cell-based model system where MTA levels could be titrated across a wide range would allow for a more direct correlation between MTA concentration and compound potency. Having demonstrated sensitivity to SAM and MTA, we leveraged CBH-002 to quantify the impact of intracellular MTA levels on PRMT5 target occupancy. This represents a unique application of BRET technology to directly probe the influence of endogenous metabolites facilitating specific drug-target interactions. In particular, the cohort of PRMT5 inhibitors should demonstrate distinct patterns of MTA sensitivity. Ligands for the cosubstrate pocket should demonstrate direct competition and mutual exclusivity with MTA. Likewise, peptide-substrate pocket, SAM-uncompetitive inhibitors should demonstrate allosteric competition with MTA, and target engagement should be weakened in the presence of saturating MTA levels. Conversely, MTA-cooperative inhibitors (i.e., inhibitors that form a ternary complex with PRMT5:MTA) should demonstrate enhanced binding with increasing MTA levels. A fixed, quantitative concentration of 30 nM BRET probe was selected to minimize the impact on occupancy of MTA and test compound. Strikingly, the influence of MTA levels on potency for each subtype varied widely (Fig. 4A–F, Fig. S5, Table S2). Additionally, all inhibitors except for GSK3326595 and PF-06939999 had Hill slopes significantly less than 1, some of which then approached 1 as [MTA] increased (Table S2). PRMT5 is reported to exist predominantly as a heterooctamer with its partner protein WDR77 (MEP50)[11] and we speculate that CBH-002 can report on crosstalk between binding to PRMT5 within the multimeric complex. Consequently, the Hill slope was allowed to vary when fitting the initial concentration response data for all compounds.

For compounds known to bind to the cosubstrate binding site (cosubstrate-competitive), or to the peptide-substrate binding site in the presence of SAM (SAM-uncompetitive), increasing concentrations of MTA resulted in a decrease in the affinity of the inhibitors for PRMT5 and a right-shift in the concentration-response curves. Cosubstrate-competitive inhibitors such as LLY-283 and PF-06939999 demonstrated right-shifted pharmacology with increasing exogenous MTA, where addition of 300 μM MTA caused a 6-fold right-shift in the binding of LLY-283 and a more modest 3-fold shift in affinity of PF-06939999 (Fig. 4A, B, Fig. S5A, B, Table S2), consistent with a competitive binding mode of action. Although LLY-283 is known to bind the cofactor pocket, consistent with the mode of action revealed here, our data conflicts with results from a biochemical SAM-competition assay which did not show competitive binding[8]. The SAM-uncompetitive inhibitors EPZ015666 and GSK3326595 demonstrated modest right-shifts in potency upon treatment of HEK293 cells with exogenous MTA (3–4-fold at 300 μM exogenous MTA) (Fig. 4C, D, Fig. S5C, D, Table S2). In contrast to the first-generation inhibitors, treatment of cells with MTA increased the affinity of MRTX1719 and TNG908 for PRMT5 by one or more orders of magnitude (Fig. 4E, F, Fig. S5E, F, Table S2). While maximal potency shift was relatively similar

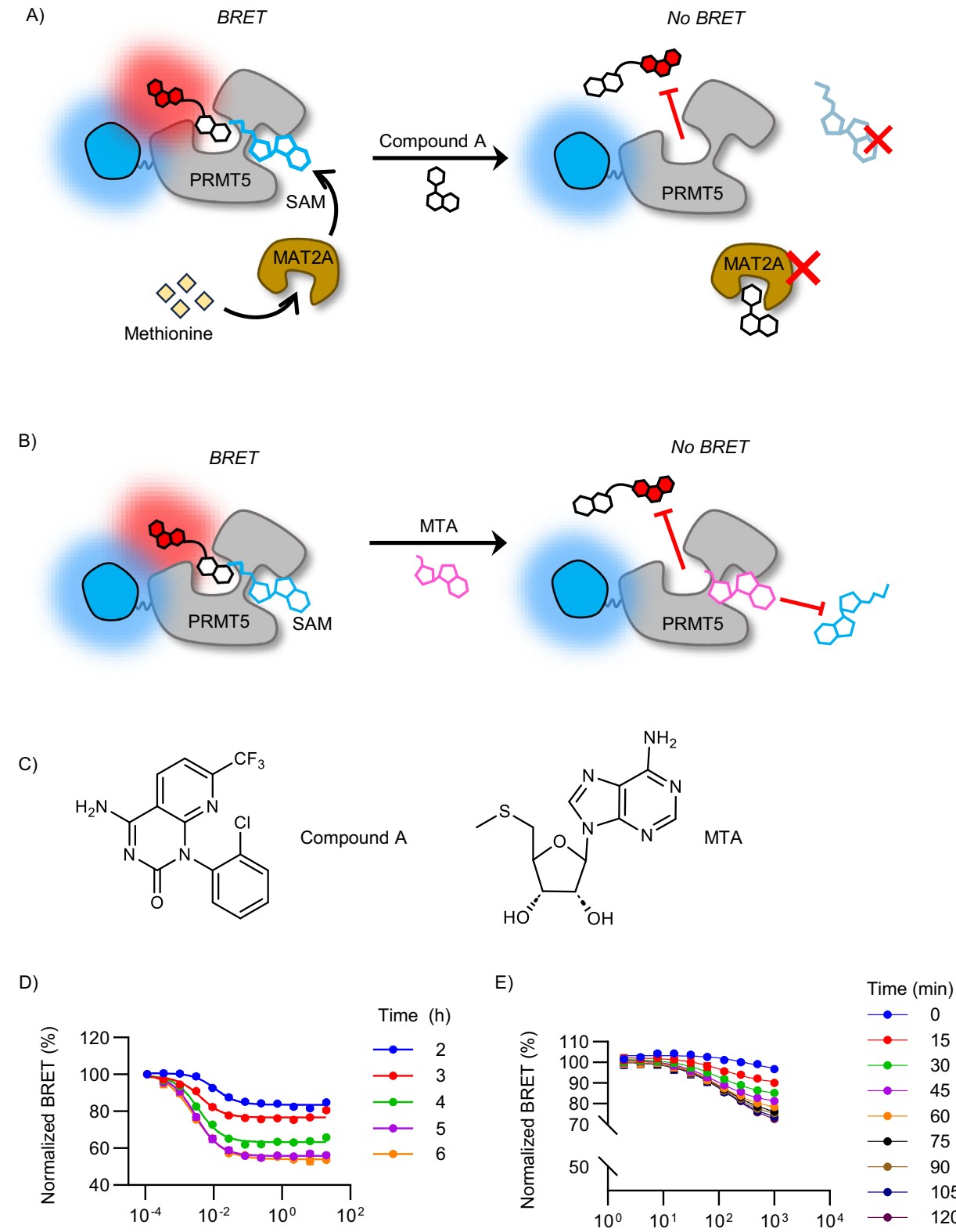

**Fig. 3 | BRET probe CBH-002 is sensitive to intracellular SAM and MTA levels.**
**A** Experimental schematic for testing BRET probe sensitivity to SAM. Intracellular SAM production is regulated by the MAT2A enzyme, thus reducing SAM levels via MAT2A inhibition should lead to BRET probe displacement. **B** Experimental schematic for testing BRET probe sensitivity to MTA. If CBH-002 behaves similarly to parental compound GSK3326595, it should bind more favorably to SAM:PRMT5. Thus, treating cells with MTA should displace both SAM and the BRET probe. **C** Schematics for MTA and Compound A (MAT2A inhibitor) used in these experiments. **D** Representative data showing BRET probe signal as a function of MAT2A

inhibition and time at 37 °C. HEK293 cells expressing NLuc-PRMT5 were pre-treated with CBH-002 for 2 h prior to treatment with a titration series of MAT2A inhibitor (Compound A). The plate was read in 1 h intervals at a temperature of 37 °C. Data are the mean ± SEM of 3 independent experiments (n = 3). **E** Representative data showing BRET probe signal as a function of MTA concentration at room temperature. HEK293 cells expressing NLuc-PRMT5 were pre-treated with BRET probe CBH-002 for 2 h prior to treatment with a titration series of MTA. The plate was read in 15 min intervals at room temperature. Data are the mean ± SEM of 3 independent experiments (n = 3). Source data are provided as a Source Data file.

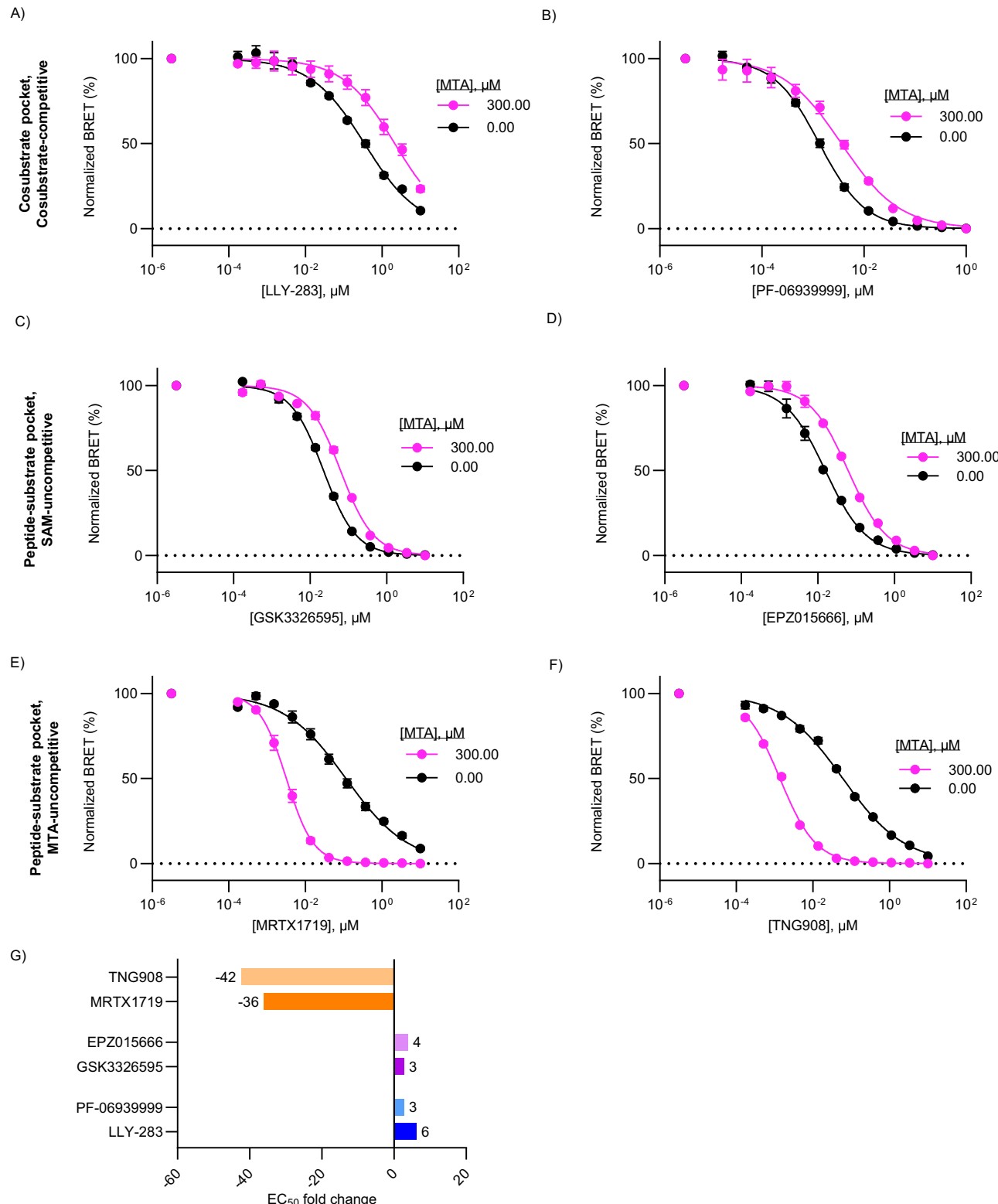

**Fig. 4 | Influence of MTA on PRMT5 inhibitor engagement. A–F** Representative compound concentration curves for PRMT5 inhibitors under low and high MTA concentration. HEK293 cells were treated with 30 nM CBH-002, a 300 μM, 7 pt, 3-fold series of MTA, and an 11 pt, 3-fold series of PRMT5 inhibitor for 2 h prior to BRET measurements. Vehicle treatment and 300 μM MTA conditions shown (see Fig. S4 for full dataset). Data are the mean ± SEM of 3 independent experiments (n = 3). Cofactor competitive (**A**, **B**) and SAM-uncompetitive (**C**, **D**) inhibitors exhibit weakened affinity in the presence of high [MTA]. MTA-cooperative inhibitors (**E**, **F**) exhibit enhanced potency in the presence of high [MTA]. **G** $EC_{50}$ shift fold-change for PRMT5 inhibitors. Values were calculated as the difference between compound $EC_{50}$ at 300 μM MTA and compound $EC_{50}$ with vehicle treatment. Negative values represent compounds which experienced enhanced potency under high MTA conditions. Source data are provided as a Source Data file.

between the two compounds (35-fold vs 43-fold for MRTX1719 or TNG908, respectively), MRTX1719 exhibited higher sensitivity compared to TNG908 at lower MTA concentrations (Table S2). To provide further information on the mechanism of PRMT5 engagement, the impact of MTA on compound binding was analyzed by global fitting of the data. For the cosubstrate competitive (LLY-283 and PF-06939999) and SAM-uncompetitive inhibitors (GSK3326595 and EPZ015666), where addition of MTA reduces compound affinity, the data were fit to the Gaddum/Schild $EC_{50}$ equation using GraphPad Prism (Fig. S5G, Table S3)[38]. In each case, the $EC_{50}$ value derived from global fitting was close to the value at [MTA] = 0 in the individual data set analysis. In addition, the Schild slope was close to 1, consistent with a binding model in which MTA acts competitively to displace the compound. Subsequent fitting in which the Schild slope was constrained to 1 provided $K_b$, the apparent affinity of the complex for MTA, with values ranging from ~50 to 170 μM. For MRTX1719 and TNG908, MTA increased the affinity of the inhibitor for the enzyme and also resulted in a change in Hill slope from ~0.5 to ≥1. We speculated that the increase in Hill slope at high [MTA] resulted in $EC_{50}$ values that approached the enzyme concentration requiring an analysis that accounted for tight binding. As discussed above, we ascribed the Hill slope of 0.5 to negative cooperativity between two binding sites in the multimeric protein complex. Therefore, we globally analyzed the data using a model comprised of two quadratic functions each based on the Morrison equation (Fig. S5H)[39]. Fitting of the MRTX1719 MTA titration data yielded an enzyme concentration of 1.6 nM and values of $K_i$ and $K_{ii}$ of 1 μM and 13 nM at [MTA] = 0 that decreased to 2.6 and 0.4 nM at high MTA. A similar trend was observed for TNG908 confirming that the affinity of the MTA uncompetitive inhibitors for the two PRMT5 binding sites increased dramatically in the presence of MTA (30–400-fold). Thus, our results support an MTA-uncompetitive mode of target engagement for both MRTX1719 and TNG908 at PRMT5. While "MTA-cooperative" is used to define inhibitors whose affinity increases when MTA is bound, we would like to suggest that "MTA-uncompetitive" is a more precise definition for the mode of action of these compounds since cooperativity can in principle be either positive or negative. The enhanced potency of MRTX1719 in the presence of saturating MTA also supports the potential for protracted drug residence time at MTA-bound PRMT5 complexes, compared to native PRMT5 in a wild-type setting. To query the influence of [MTA] on MRTX1719 residence time, we performed washout experiments in live cells under conditions of low and saturating [MTA]. In the absence of exogenous MTA, a near-saturating dose of MRTX1719 demonstrated slow, but measurable dissociation kinetics over 6 h of analysis (Fig. S5I). In contrast, at saturating [MTA], intracellular drug residence time was further enhanced, with only fractional drug dissociation within this timeframe. Together these results support that MTA-uncompetitive inhibitors demonstrate enhanced durability and potency at PRMT5 as a function of MTA occupancy.

**Surveying synthetic lethal engagement of PRMT5-MTA complexes in MTAP-knockout cells**

MTA-uncompetitive inhibitors, exemplified by MRTX1719 and TNG908, demonstrate a wide range of potencies across MTAP-deleted cancer cell lines[24]. Intrigued by the possibility that the intracellular SAM/MTA pool may be variable among such cancer cells, and that this variability could be a driver in drug potency in various cell types, we measured target engagement for the inhibitors in a more pathophysiological setting. HCT116 MTAP WT and MTAP$^{-/-}$ colorectal cancer cells constitute an isogenic pair which has served as the model system to benchmark synthetic lethal PRMT5 inhibitors. Previous studies have demonstrated a selective antiproliferative effect in HCT116 MTAP$^{-/-}$ cells for MTA-uncompetitive PRMT5 inhibitors as well as MAT2A inhibitors, supporting synthetic lethality via both nodes within the SAM/PRMT5 pathway[19–22,40]. In contrast, SAM-uncompetitive inhibitors

such as GSK3326595 showed little discrimination between the HCT116 MTAP$^{-/-}$ versus wild-type isogenic pair, suggesting that the elevated MTA levels uniquely potentiate ternary complex formation with MTA-uncompetitive inhibitors[20,21]. Similar to HEK293 cells, expression of NLuc-PRMT5 DNA alone in HCT116 wild-type cells resulted in biphasic BRET dose-response curves, which could be resolved by co-expression with WDR77 (Fig. 5A, B, S6A–D). In the HCT116 isogenic pair, CBH-002 potency was comparable to that of HEK293, indicating that intracellular SAM levels are similar across these cell models (Fig. 5A, Fig. S6C). These results align with earlier reports suggesting that deletion of MTAP does not significantly affect intracellular SAM homeostasis, despite alterations in MTA[41]. Next, we investigated target engagement of SAM- and MTA-uncompetitive inhibitors in HCT116 MTAP wild-type and -deleted cells. Consistent with our previous MTA titration experiments, as well as the equipotent antiproliferative effects observed for GSK3326595 in the HCT116 MTAP isogenic pair, affinity of GSK3326595 was not affected by MTAP status (Fig. 5C, S6E, Table S4). In contrast, MRTX1719 demonstrated a strong increase in engagement in MTAP$^{-/-}$ HCT116 cells ($EC_{50}$ = 5.9 nM) compared to wild-type cells ($EC_{50}$ > 1 μM) (Fig. 5C, S6F, Table S4). Notably, these results are in good agreement with previous phenotypic analyses in HCT116 isogenic pairs (Table S1), demonstrating that target engagement of MTA-uncompetitive inhibitors is commensurate with apparent antiproliferative effects in such cell types. Our data further support the molecular mechanism of action for these inhibitors in a more physiologically relevant setting, and the use of CBH-002 to query synthetic lethal, ternary complex inhibitors of PRMT5 in cells.

It is noteworthy that the potency of MRTX1719 in wild-type HCT116 colorectal cancer cells was orders of magnitude weaker than that observed in HEK293 cells (>1 μM vs 114 nM), indicating that intracellular [MTA] may be significantly lower in HCT116 cells. To investigate MTA-sensitivity to MRTX1719 we performed MTA titration experiments in wild-type HCT116. In contrast to our previous results using HEK293, MRTX1719 engagement in HCT116 cells revealed a far broader range of sensitivity to exogenous MTA, spanning over two orders of magnitude (Fig. 5D, E, Table S5). Remarkably, at single-digit micromolar MTA concentrations, MRTX1719 potency was equivalent to that observed in MTAP$^{-/-}$ HCT116 cells. Thus, MRTX1719 potency can be used to estimate the relative intracellular [MTA] across cell lines. These results suggest that [MTA] concentrations in MTAP$^{-/-}$ HCT116 cells accumulate to the low micromolar range, whereas levels in wild-type HCT116 cells are at least two orders of magnitude lower. The fact that MRTX1719 exhibits lower $EC_{50}$ values in HEK293 versus HCT116 cells suggests that MTA homeostasis may also vary in a wild-type MTAP setting.

## Discussion

PRMT5 is an established anti-cancer target for which various classes of inhibitors have been developed. These drugs exploit either the cosubstrate or substrate binding pocket comprising PRMT5-SAM complex binders that are competitive with the substrate (SAM-uncompetitive, e.g., EPZ015666 and GSK3326595) and inhibitors such as LLY-283 and PF-06939999, which bind to the cosubstrate binding site and are competitive with SAM/MTA (cosubstrate-competitive). In addition, a third class of inhibitors has recently been reported, including MRTX1719 and TNG908, that preferentially bind to the peptide-substrate pocket when MTA is bound in the cosubstrate pocket (MTA-uncompetitive). In 2021, an SPR-based fragment screen yielded the first described compound selective for the PRMT5-MTA complex[24]. Following hit optimization, MRTX1719 was characterized as a peptide-substrate-competitive, MTA-cooperative inhibitor with promising selectivity for MTAP-deleted cancer cells[21]. The discovery of MTA-uncompetitive PRMT5 inhibitors has galvanized drug discovery efforts to generate precision cancer medicines since MTA is elevated in many cancers due to deletion of MTAP, thereby exposing a synthetic

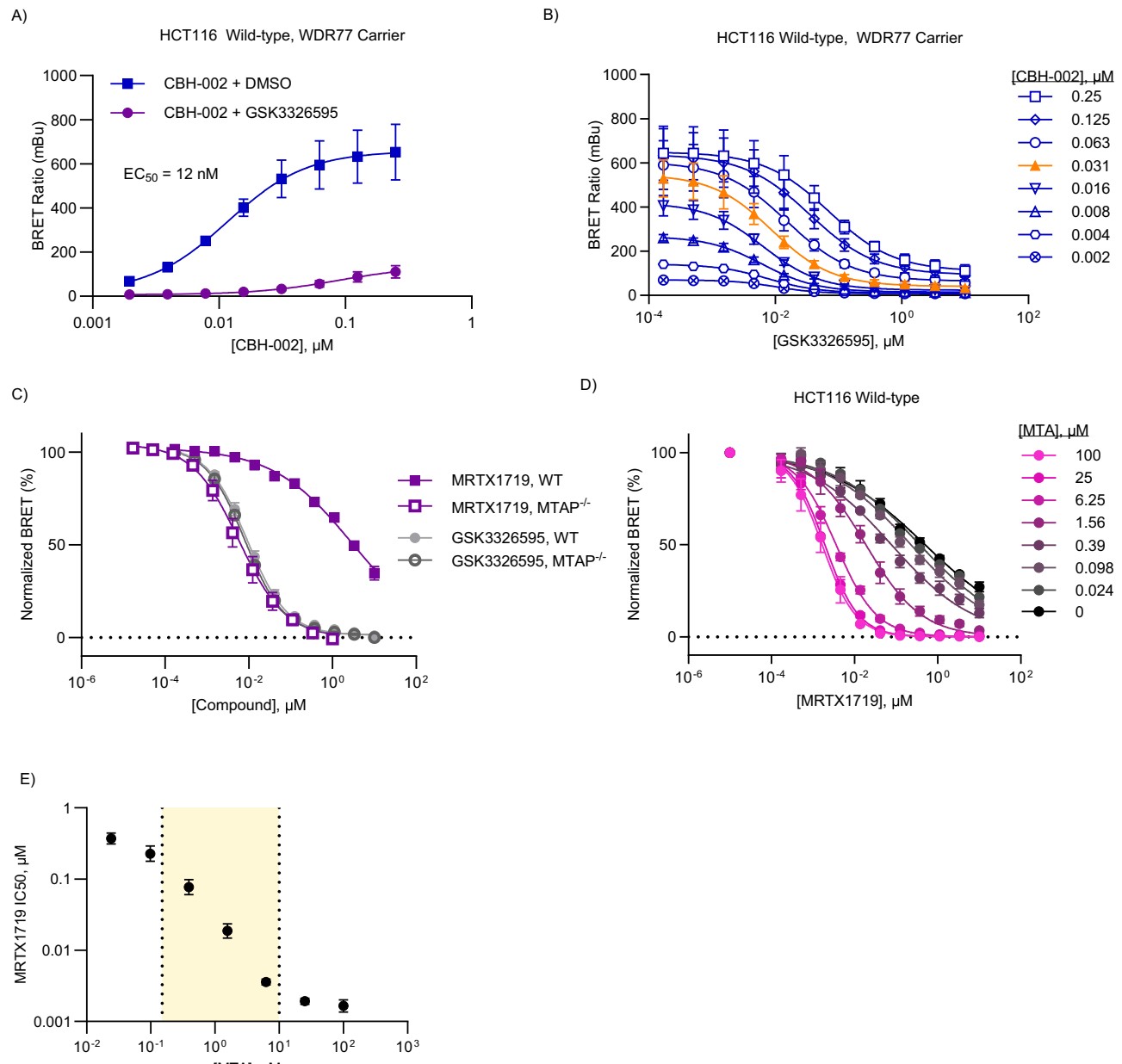

**Fig. 5 | Surveying synthetic lethal engagement of PRMT5-MTA complexes in MTAP-knockout cells. A, B** BRET probe CBH-002 validation data in HCT116 wild-type cells with WDR77 co-expression. HCT116 wild-type cells expressing NLuc-PRMT5 and unlabeled WDR77 DNA were treated with an 11 pt 3-fold series of GSK3326595 and an 8 pt, 2-fold titration series of CBH-002 for 2 h before BRET was measured. Data are the mean ± SEM of 3 independent experiments (n = 3). Data was fit with 4-parameter regression curve (see "Methods"). Data is plotted as both an affinity curve (**A**) and compound matrix (**B**). Orange triangles represent the optimal BRET probe concentration. **C** Representative compound concentration curves for GSK3326595 and MRTX1719 in HCT116 MTAP wild-type and MTAP⁻/⁻ cells. HCT116 cells co-expressing NLuc-PRMT5 and untagged WDR77 DNA were treated with 11 pt, 3-fold titrations of compound and an 8 pt 2-fold series of CBH-002 for 2 h prior to BRET measurement. The curves at optimal probe concentration (30 nM) are shown (for full matrices, see Fig. 5B, S6D–F). Curves were normalized and fit with a 4-parameter regression model (see "Methods"). Data are the mean ± SEM of 3 independent experiments (n = 3). **D** Titration matrix of MTA and MRTX1719 in HCT116 MTAP wild-type cells. HCT116 MTAP wild-type cells were treated with 30 nM CBH-002, and a matrix of MTA and MRTX1719 for 2 h before BRET was measured. Data were normalized and fit with a 2-parameter regression model (see "Methods"). Data are the mean ± SEM of 3 independent experiments (n = 3). **E** EC₅₀ measurements from **D** replotted as a function of MTA concentration. Yellow shaded area indicates linear range of quantitation. Data are the mean ±95% CI. Source data are provided as a Source Data file.

lethal vulnerability. Driven by the knowledge that inhibition of PRMT5 is modulated both by interaction of PRMT5 with partner proteins such as WDR77 and the levels of metabolites such as SAM and MTA, we set out to develop a method for assessing the pharmacological impact of PRMT5 inhibitors directly within live cells. Here, we address this important gap, by significantly expanding the portfolio of cellular PRMT5 TE and global selectivity assays with chemoproteomic and 2D thermal profiling, as well as quantitative live-cell NanoBRET enabled by

CBH-002, a BRET biosensor that can quantify PRMT5 engagement by all three classes of PRMT5 inhibitors across a broad range of intracellular SAM and MTA concentrations. Previously, cellular target engagement for PRMT5 inhibitors has been demonstrated via western blot-based detection of symmetrically dimethylated nuclear proteins[42,43], as well as via a PRMT5-RIOK1 protein-protein interaction NanoBiT assay in permeabilized cells[44,45], and functional end-point assays monitoring cell viability (see Table S1). We based our BRET

probe design on EPZ015666, a highly potent and clinically evaluated PRMT5 inhibitor targeting the substrate pocket. Due to its binding mode within the peptide-binding groove adjacent to the SAM pocket, CBH-002 is sensitive towards any inhibitors that affect substrate or cosubstrate binding. Cellular binding potencies for the SAM-uncompetitive (GSK3326595 and EPZ015666) and SAM-competitive (LLY-283 and PF-06939999) compounds agreed well with literature values, and addition of cell-permeable MTA resulted in a right-shift in the concentration-response curves, indicating a decrease in potency for PRMT5 due to the displacement of SAM from the enzyme by MTA. Reportedly, while the biochemical $IC_{50}$ value for inhibition of PRMT5 by MRTX1719 was only enhanced 5-fold in the presence of MTA, cell-based assays based on biomarkers of PRMT5 activity or cell viability assays demonstrated a more marked enhancement of MRTX1719 activity, showing an 82-fold selectivity in MTAP-deleted versus MTAP wild-type isogenic pairs[20]. We quantified the binding of the MTA-uncompetitive inhibitors MRTX1719 and TNG908 to PRMT5 in HEK293 cells and observed $EC_{50}$ values of 60 and 114 nM, respectively, which are intermediate between reported biochemical (low nM) and cellular (μM) potencies for these inhibitors, and which we speculate indicates that HEK293 cells contain sufficient MTA to form the PRMT5-MTA complex. Again, in agreement with the known binding mode, addition of MTA resulted in a left-shift in the concentration-response curves indicating an increase in affinity for PRMT5. Direct binding of MRTX1719 to PRMT5 in the HCT116 MTAP-deletion isogenic pair was performed to assess the impact of MTA levels on PRMT5 inhibition in a cell line that is directly relevant to the synthetic lethal phenotype. As expected, we observed that binding of MRTX1719 to PRMT5 was more potent in HCT116 MTAP[-/-] cells compared to wild-type cells. However, while we only observed a ~35-fold increase in affinity of PRMT5 in HEK293 cells with 300 μM MTA, the difference in affinity in the isogenic pair was greater than 1,000-fold, ($EC_{50}$ = 5.9 nM and $EC_{50}$ > 1 μM for MTAP[-/-] and MTAP wild-type, respectively). Intriguingly, viability and pathway analysis metrics indicate an enhancement in selectivity of ~100-fold for MRTX1719 in HCT116 MTAP[-/-] cells over wild-type cells[20,21], suggesting that the BRET system may enable a more physiologically accurate assessment of pharmacology for uncompetitive PRMT5-MTA complex inhibitors and could serve as a screening tool to identify novel synthetic lethal inhibitors[46].

Collectively, our data confirm that CBH-002 can quantify the binding of inhibitors to PRMT5 in live cells, including MTA-uncompetitive compounds, and is sensitive to metabolite levels. We hypothesize the versatility of CBH-002 is linked to the binding mode of the tetrahydroisoquinoline scaffold, which can sense changes in substrate- and cosubstrate binding. Our results further suggest that structurally diverse BRET probe development for PRMT5 may be feasible using MTA-uncompetitive inhibitors such as MRTX1719 as a parent scaffold. Nonetheless, the slow binding kinetics observed for MRTX1719 may restrict its utility in accurately measuring target engagement. Ideally, such BRET biosensors could be leveraged to not only quantify uncompetitive PRMT5 engagement, but also gauge the local intracellular concentrations of the key metabolites mediating drug cooperativity. Such knowledge will be critical in the design of new medicines, as the intracellular SAM and MTA concentrations influence the vulnerability of metabolite-cooperative inhibitors. Previous studies have attempted to estimate the free intracellular and extracellular concentrations of these metabolites[47]. For example, in MTAP null glioblastoma, free MTA was predominantly secreted into culture medium, vastly exceeding the measured intracellular pool. In contrast to these spectroscopic studies of free metabolites, our PRMT5 BRET biosensor may offer a complementary approach to estimate the relative concentrations of MTA and SAM bound by PRMT5 within cells. As such, the biosensor could serve to stratify the relative levels of PRMT5-bound SAM and MTA across various normal- and cancer cell populations. Within live cells and also cell-free

conditions, we observed exquisite sensitivity of CBH-002 to [SAM], supporting use of the BRET biosensor to gauge relative intracellular [SAM] levels across cell lineages. CBH-002 potencies were similar across the cell lines used for our studies, supporting that SAM levels are fairly uniform in HCT116 and HEK293 cells. These results are consistent with previous reports indicating that SAM levels are maintained consistently in most cell culture experiments, and generally unaffected by MTAP status[34,41]. However, CBH-002 potency was less sensitive to [MTA], and could not be used to evaluate subtle fluctuations at lower concentrations of intracellular MTA. To better estimate relative MTA concentrations across cell lines, MRTX1719 sensitivity was established over nearly two orders of magnitude in HCT116 wild-type cells. In contrast to the SAM pool, the potency of MRTX1719 was highly variable in HCT116 wild-type, HCT116 MTAP[-/-], and HEK293, supporting highly disparate pools of MTA in these cell lines. It may be therefore valuable to perform similar studies in various normal- and cancer cell lineages to determine if the variability in these key metabolites may underlie selectivity of MTA-cooperative inhibitors in diverse cancer cell populations. If such relationships could be established, a PRMT5 BRET biosensor like CBH-002 could serve to stratify the therapeutic potential of PRMT5 inhibitors as precision medicines. In summary, while previous methods have provided valuable insights into methyltransferase inhibition, our development of a PRMT5 BRET biosensor represents a significant advancement by enabling the direct measurement of MTA-uncompetitive target engagement in live cells, a capability not afforded by earlier BRET approaches focused on competitive binding across broader target classes such as kinases. Thus, we envisage our assay will be valuable for the design, evaluation, and optimization of next generation PRMT5 MTA-uncompetitive inhibitors (Fig. 6). This unique methodology for directly assessing metabolite-dependent drug-target engagement in live cells has the potential to significantly enhance our understanding of drug mechanisms and accelerate the development of precision medicines, particularly in contexts where metabolite levels play a critical role in drug sensitivity, such as in MTAP-deleted cancers.

## Methods
General synthetic procedures are described in the Supplementary Information.

### Chemical proteomics
Chemical proteomic experiments with GSK3326595 were conducted as previously described[48]. In brief, KMS-11 were cultured at 37 °C, in a humidified 5% $CO_2$ atmosphere in RPMI-1640 (Life Technologies) + 10% FBS. To obtain lysate, the cells were harvested at 80 % confluency from five T75 flasks (Greiner) and subsequently pelleted and washed with DPBS. The obtained pellets were then lysed by addition of 3x pellet volume of lysis buffer (50 mM Tris pH 7.5, 0.8% v/v NP-40, 5% v/v glycerol, 1.5 mM $MgCl_2$, 100 mM NaCl, 25 mM NaF, 1 mM $Na_3VO_4$, 1 mM PMSF, 1 mM DTT, 10 μg/mL TLCK, 1 μg/mL leupeptin, 1 μg/mL aprotinin, 1 μg/mL soybean trypsin) on ice, before being pushed gently 10-times through a 21 G needle (Braun) using a 1 mL syringe (Braun, Inject-F). The lysates were allowed to incubate on ice for 10 min before being centrifuged at 17,000 × $g$ at 4 °C for 30 min. Next, total protein concentration was adjusted to 10 mg/mL and the lysates were stored for further use at −80 °C. In preparation for a pulldown, the amine derivatized compound CBH-001 was coupled to NHS-activated Sepharose 4 Fast Flow beads (Cytiva). 100 μL beads as a slurry in 50% isopropanol were used for each experiment, the experiments were done in triplicates (with/without competition). The beads were washed with 500 μL DMSO and centrifuged at 0.1 × $g$ at RT for 3 min. The supernatant was removed, and the washing step was repeated 3×. Then the bead bed was re-suspended in 50 μL DMSO. To immobilize the couplable compound on the beads, 2 μL of a 10 mM stock in DMSO was added to the

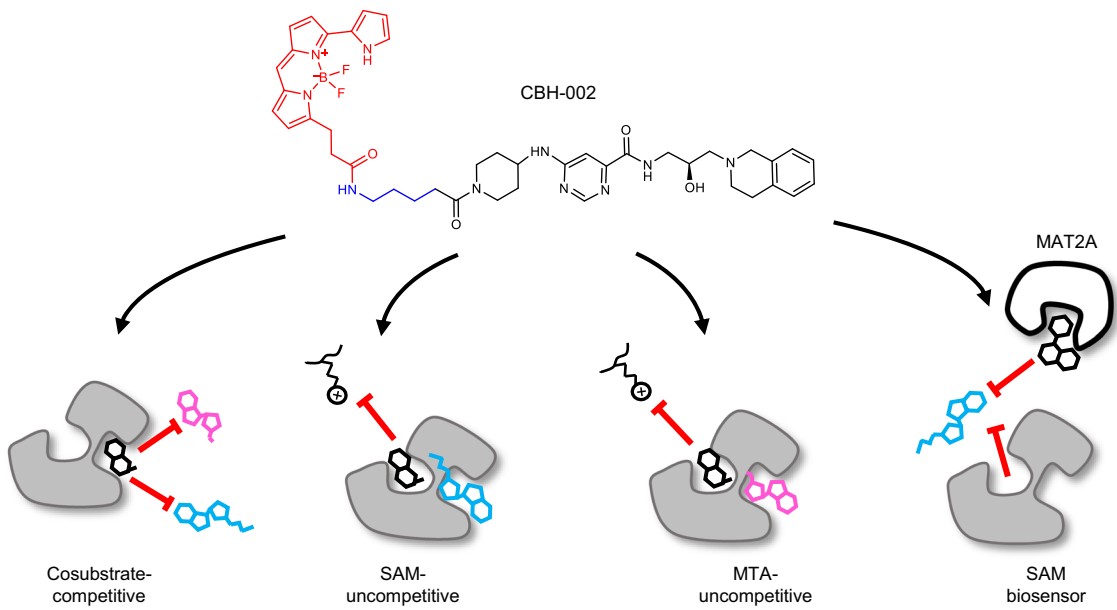

**Fig. 6 | Schematic displaying versatility of the PRMT5 BRET probe CBH-002.** CBH-002 can sense cosubstrate competitive, SAM- and MTA-uncompetitive modes of action and act as a SAM biosensor.

beads along with 0.75 μL DIPEA (diisopropylethylamine). Then the tubes were end-to-end rotated on a wheel for 20 h at RT. Subsequently, successful immobilization and depletion of the free amine from the supernatant was determined by LC-MS analysis. The remaining unreacted NHS (N-hydroxysuccinimide) groups were blocked with 2.5 μL ethanolamine and end-to-end rotated on a wheel for 8 h at RT. The lysates were thawed on ice before pre-treatment with DMSO control (n = 3) or parent compound, GSK3326595 (n = 3), at 10 mM and incubating for 30 min at 4 °C. A total cell lysate (TCL) sample was kept aside (n = 1). The derivatized beads were washed 3× with 500 μL DMSO and then 3× with freshly prepared lysis buffer by centrifugation at 0.1 × g for 3 min at RT and subsequent removal of the supernatant. After that, the derivatized beads were combined with the cell lysate at 1.6 mg of protein per pull-down (10 mg/mL) being either compound or DMSO control pre-treated in triplicates. Beads and lysates were then end-to-end rotated on a wheel in the cold room at 4 °C for 2 h. Then the samples were centrifuged at 0.1 × g for 3 min at 4 °C and 300 μL of the supernatant was removed. Working in the cold room at 4 °C, the slurry was re-suspended in lysis buffer and 500 μL were transferred to a Bio Spin column (Bio-Rad) and the beads were allowed to settle by gravity. Next, the columns were washed with 5 mL lysis buffer via gravity flow and centrifuged at 0.1 × g for 1 min at 4 °C to the remove the remaining supernatant. Then the protein was eluted for Western Blot experiments using 100 μL of 2× sample buffer (65.8 mM Tris-HCl pH 6.8, 26.3% (w/v) glycerol, 2.0% SDS, 0.01% bromophenol blue, 50 mM DTT) and boiling the beads at 100 °C for 10 min. The samples were stored in the freezer at −20 °C.

### Western blot
The eluted proteins were analyzed using a polyacrylamide gel and 1x MES buffer (50 mM MES, 50 mM Tris Base, 0.1% SDS, 1 mM EDTA, pH 7.3) and transferal to a nitrocellulose blotting membrane. The membrane was blocked with blocking buffer (2.5% m/v) BLOT-Quick Blocker (Merck) in PBST (Phosphate-buffered saline with Tween: 4.3 mM Na$_2$HPO$_4$, 1.47 mM KH$_2$PO$_4$, 137 mM NaCl, 2.7 mM KCl, 0.05% (v/v) Tween 20) on a shaker in the dark for 1 h at RT. Then the blot was probed with the antibodies EPR5772 (Abcam, ab109451, 1:10000), MEP50 (2823S, Cell Signaling Tech., 1:1000) and G-9 (s-365062, Santa Cruz, 1:200) and imaged at the Odyssey CLx (Li-cor) at wavelength 700 and 800 nm.

### Mass spectrometry sample preparation
100 μL of each sample were combined 1:1 with freshly prepared 0.1 M TRIS solution. Then 5 μL 200 mM DTT were added and the mixture incubated for 1 h at RT. Following subsequent alkylation with 20 μL 200 mM iodoacetamide samples were incubated for 30 min in the dark at RT. The samples were diluted to 300 μL with TEAB and incubated at 37 °C overnight. To precipitate the proteins, 600 μL MeOH and 150 μL chloroform were added to each vial and after vortexing 450 μL milliQ-water were added. After centrifugation at 15,000 × g for 5 min at RT, the upper phase was gently pipetted off without touching the precipitate at the interface. Next, 450 μL of MeOH were added, and the upper phase was pipetted off gently again. This was repeated once and then the samples were centrifuged at top speed. The supernatant was removed the vials left to dry 60 min RT. The precipitate was resuspended in 50 μL 6 M urea buffer, followed by vortexing and sonication for 5 min. The solution was diluted with 250 μL milliQ-water. Then trypsin in a 1:50 ratio regarding the total protein content was added, followed by incubation at 37 °C overnight. A SEP-PAK C18 purification was performed using solution A (98% milliQ-water, 2% acetonitrile, 0.1% formic acid) and solution B (35% milliQ-water, 65% acetonitrile, 0.1% formic acid). The samples were acidified with 1% formic acid. Sola HRP SPE cartridge (Thermo Fisher) were attached to a vacuum manifold. First, the columns were equilibrated with 500 μL solution B. Second, 1000 μL of solution A were put on the column and again a small supernatant was left to prevent the column from running dry. Third, the peptide digest sample was loaded on the column. After that, the column was washed again with 1000 μL solution A. The columns were placed after the washing in fresh 1.5 mL Eppendorf vials. Then 600 μL solution B was used to elute into a fresh vial. Then, the samples were dried in the SpeedVac vacuum concentrator for 24 h. Mass spectrometry data were acquired at the Discovery Proteomics Facility (University of Oxford). Peptides were resuspended in 5% formic acid and 5% DMSO and then trapped on an Acclaim™ PepMap™ 100 C18 HPLC Columns (5 μm × 0.1 mm × 20 mm, Thermo Fisher Scientific) using solvent A (0.1% formic acid in water) at a pressure of 60 bar and separated on an Ultimate 3000 UHPLC system (Thermo Fischer Scientific) coupled to a QExactive mass spectrometer (Thermo Fischer Scientific). The peptides were separated on an Easy Spray PepMap RSLC column (75 μm i.d. ×2 μm × 50 mm, 100 Å, Thermo Fisher) and then electro sprayed directly into an QExactive mass spectrometer

(Thermo Fisher Scientific) through an EASY-Spray nano-electrospray ion source (Thermo Fisher Scientific) using a linear gradient (length: 60 min, 5–35% solvent B (0.1% formic acid in acetonitrile), flow rate: 250 nL/min). The raw data was acquired on the mass spectrometer in a data-dependent mode (DDA). Full scan MS spectra were acquired in the Orbitrap (scan range 380–1800 m/z, resolution 70000, AGC target $3e^6$, maximum injection time 100 ms). After the MS scans, the 15 most intense peaks were selected for HCD fragmentation at 28% of normalized collision energy. HCD spectra were also acquired in the Orbitrap (resolution 17500, AGC target $1e^5$, maximum injection time 128 ms) with first fixed mass at 100 m/z.

## MS data analysis

Raw data was processed using MaxQuant version 1.6.1.2 and the reference complete human proteome FASTA file (UniProt). Label-Free Quantification (LFQ) and Match Between Runs were selected; replicates were collated into parameter groups to ensure matching between replicates only. Cysteine carbamidomethylation was selected as a fixed modification and methionine oxidation as a variable modification.

group. Each aliquot was heated to different temperatures for 3 min in a PCR machine (Bio-Rad), with the temperature range set to 42–64 °C and 2-degree intervals. Cell pellets were then lysed in 0.1% NP-40 Tris-NaCl lysis supplemented with protease and phosphatase inhibitors (Sigma). Cell lysis was facilitated by three cycles of rapid freeze-thawing in liquid nitrogen. The lysates were clarified by centrifugation at $17,000 \times g$ for 20 min, and BCA assay was performed on the soluble fraction. For MS analysis, 100 μg of protein per aliquot was taken.

## MS analysis of TP samples

To reduce and alkylate proteins, lysate was first incubated with DTT up to a final concentration of 5 mM for 1 h at RT followed by a 1 h incubation with iodoacetamide added to a final concentration of 20 mM. The proteins were then acetone-precipitated overnight −20 °C and pelleted at $8000 \times g$ for 10 min at 4 °C. Dry pellets were resuspended in 100 μL of 50 mM TEAB and 2.5 μg of Trypsin/LysC (Promega) was added for overnight digestion at 37 °C. TMT labeling (Thermo Fisher) was performed according to the manufacturer's protocol, and the samples were pooled according to the following scheme:

| Temp (°C) | 126 | 127L | 127H | 128L | 128H | 129L | 129H | 130L | 130H | 131 | RefCol | | |
|---|---|---|---|---|---|---|---|---|---|---|---|---|---|
| 42 | 5 | 1 | 0.2 | 0.04 | 0 - | - | - | - | - | | 128H | F1 | |
| 62 | - | - | - | - | - | 5 | 1 | 0.2 | 0.04 | 0 | 131 | | |
| 44 | 5 | 1 | 0.2 | 0.04 | 0 - | - | - | - | - | | 128H | F2 | |
| 64 | - | - | - | - | - | 5 | 1 | 0.2 | 0.04 | 0 | 131 | | |
| 46 | 5 | 1 | 0.2 | 0.04 | 0 - | - | - | - | - | | 128H | F3 | |
| 58 | - | - | - | - | - | 5 | 1 | 0.2 | 0.04 | 0 | 131 | | |
| 48 | 5 | 1 | 0.2 | 0.04 | 0 - | - | - | - | - | | 128H | F4 | |
| 60 | - | - | - | - | - | 5 | 1 | 0.2 | 0.04 | 0 | 131 | | |
| 50 | 5 | 1 | 0.2 | 0.04 | 0 - | - | - | - | - | | 128H | F5 | |
| 54 | - | - | - | - | - | 5 | 1 | 0.2 | 0.04 | 0 | 131 | | |
| 52 | 5 | 1 | 0.2 | 0.04 | 0 - | - | - | - | - | | 128H | F6 | |
| 56 | - | - | - | - | - | 5 | 1 | 0.2 | 0.04 | 0 | 131 | | |

Default settings for identification and quantification were used. Specifically, a minimum peptide length of 7, a maximum of 2 missed cleavage sites, and a maximum of 3 labeled amino acids per peptide were employed. Through selection of the 'trypsin/P' general setting, peptide bond cleavage at arginine or lysine (followed by any amino acid) was considered during in silico digest of the reference proteome. The allowed precursor and fragment ion mass tolerances were 4.5 ppm and 20 ppm, respectively. Peptides and proteins were identified utilizing a 0.01 false discovery rate, with "Unique and razor peptides" mode selected for both identification and quantification of proteins (razor peptides are uniquely assigned to protein groups and not to individual proteins). At least 2 razor + unique peptides were required for valid quantification. Processed data was further analyzed using Perseus version 1.6.2.1 and Microsoft Excel. Peptides categorized by MaxQuant as 'potential contaminants', 'only identified by site' or 'reverse' were filtered, and the LFQ intensities transformed by $\log_2$. Experimental replicates were grouped, and two valid LFQ values were required in at least one experimental group. Statistically significant competition was determined through the application of P2 tests, using a permutation-based FDR of 0.05 and an S0 of 2, and visualized in volcano plots.

## Thermal profiling

2D thermal profiling was performed according to previously described protocols[49]. Briefly, KMS-11 cells were grown until confluent in T-175 flasks (Greiner). To five flasks of confluent KMS-11 cells GSK33265995 was added at final assay concentrations of 5, 1, 0.2, 0.04 μM or equivalent volume of DMSO for 1 h. The cells were then detached using the Tryp-LE trypsin replacement enzyme (Gibco) and pelleted into 12 aliquots per

Pooled samples were subsequently desalted in C18 columns (Pierce) and the solvent removed under reduced pressure in a SpeedVac Vacuum concentrator. Dry peptide mass was then resuspended in 120 μL of 98% milliQ-$H_2O$, 2% acetonitrile, 0.1% formic acid (v/v) and fractionated using an UltiMate 3000 HPLC system (Thermo) at pH = 10 using a 60 min gradient of 0% to 90% acetonitrile in water. Fractions were then dried using a SpeedVac Vacuum concentrator and resuspended in 100 μL of 98%, milliQ-$H_2O$, 2% acetonitrile, 0.1% formic acid (v/v). MS analysis was performed on a QExactive HF mass spectrometer (Thermo Scientific) by nano-HPLC–MS/MS using a Dionex Ultimate 3000 nano HPLC with EASY spray column (75 μm × 500 mm, 2 μm particle size, Thermo Scientific) with a 60 min gradient of 2% to 35% (v/v) acetonitrile in water with 5% (v/v) DMSO and 0.1% (v/v) formic acid at a flow rate of 250 nL/min (600 bar per 40 °C column temperature). MS1 survey scans were acquired at a resolution of 60,000 at 375–1500 m/z and the 20 most abundant precursors were selected for CID fragmentation in a HCD cell. MS2 data were analyzed in Thermo Proteome Discoverer 2.1 according to the manufacturer's protocol.

## Western blot CETSA

CETSA was performed as described previously[49,50]. Briefly, KMS-11 cells were grown until confluent in T-175 flasks (Greiner). One flask was treated with 5 μM GSK33265995 in DMSO, whereas the control flask was treated with an equivalent volume of DMSO for 1 h. The cells were then harvested and aliquoted for heating to different temperatures for 3 min and lysed in NP-40 lysis buffer. The lysate was clarified by centrifugation at $17,000 \times g$ for 20 min and 30 μg of protein was used for SDS-PAGE and Western Blotting using anti-WDR77 antibody (Cell Signaling, 2823), 1:100.

The mass spectrometry proteomics data have been deposited to the ProteomeXchange Consortium via the PRIDE partner repository with the dataset identifier PXD028138.

## HEK293 cell transfections and general BRET measurements

HEK293 cells (ATCC) were cultured in DMEM (Thermo Fisher) + 10% (v/v) FBS (Seredigm), and HCT116 cells (Horizon Discovery, #PAR-034 and #R02-033) were cultured in McCoy's 5A (Thermo Fisher) + 10% FBS, with incubation in a humidified, 37 °C/5% $CO_2$ incubator. Cells were split 1–3 days prior to performing experiments and used at roughly 80% confluency. N-terminal NLuc fusion constructs were created for PRMT5 (residues 1–637) and BRD9 (residues 1–597) by inserting the full-length ORFs into pNLF1-N and pFN31K expression vectors, respectively (Promega), including a flexible Gly-Ser-Ser-Gly linker between NLuc and the protein of interest. C-terminal NLuc fusion constructs were created for PRMT7 (residues 1–692), PRMT9 (residues 1–845), RIPK2 (residues 1–540), and PRMT5 by inserting the ORFs into pNLF1-C expression vectors (Promega), including a Gly-Ser-Ser-Gly linker between the protein of interest and NLuc. Cells were transfected according to the manufacturer's protocol using FuGENE HD (Promega). NanoLuc fusion DNA was diluted into Transfection Carrier DNA (Promega) in Opti-MEM at a ratio of 1:9 (mass/mass), and FuGENE HD was added at a ratio of 1:3 (µg DNA:µL lipid). For experiments using WDR77, untagged WDR77 DNA (Promega) was used in place of Carrier DNA. Formed complexes were combined with cells resuspended to a density of $2 \times 10^5$ cells/mL in Opti-MEM + 1% FBS at a ratio of 1:20 (vol:vol). Cells were plated into a flask and incubated at 37 °C in a humidified 5% $CO_2$ atmosphere for 18–24 h, before being lifted and seeded into 96-well non-binding surface-treated plates (Corning). Cells were seeded at a density of $2 \times 10^5$ cells/mL in Opti-MEM + 1% FBS in 100 µL (96-well). The BRET probe CBH-002 was prepared first as a 100× stock in 100% DMSO (Sigma) and then diluted to a 20× stock in NanoBRET Tracer Dilution buffer (Promega). Test compounds were prepared as 1000× stocks in 100% DMSO and diluted to 10× stocks in Opti-MEM to prepare working stocks. Cells were equilibrated with the BRET probe and test compounds for 2 h at 37 °C and 5% $CO_2$, prior to BRET measurements. NanoBRET Complete 3x Substrate was prepared by combining Nano-Glo Substrate and Extracellular NanoLuc Inhibitor in Opti-MEM, according to the manufacturer's protocol (Promega). NanoBRET Complete 3× Substrate was added to cells and plates were measured on either a PheraSTAR FSX equipped with 450 nm (donor) and 610 nm (acceptor) filters, or a GloMAX Discover equipped with 450/8 nm BP (donor) and 600 nm LP (acceptor) filters, using a 0.3 s integration time.

BRET ratio values were calculated by dividing the acceptor by the donor signal, and raw BRET was converted to milli-BRET (mBRET) units (mBu) by multiplying by 1000.

$EC_{50}$ values were obtained by fitting the BRET probe and compound affinity curves to a sigmoidal four-parameter fit model in Graphpad Prism:

$$Y = Y_{min} + \frac{Y_{max} - Y_{min}}{1 + 10^{(LogEC50 - X)h}} \qquad (1)$$

where h is the Hill Slope.

When normalized BRET was used, mBRET values were normalized using

$$Normalized\ BRET(\%) = \frac{(Y - BRET_{min})}{BRET_{max} - BRET_{min}} \qquad (2)$$

where y = mBRET in the presence of test compound and BRET probe, $BRET_{min}$ = mBRET at the highest concentration (saturating) of test compound and BRET probe, and $BRET_{max}$ = mBRET in the presence of vehicle (DMSO) and BRET probe.

Normalized concentration-response curves showing full competition were fit using a 2-parameter model:

$$Y = \frac{100}{1 + 10^{(LogEC50 - X)h}} \qquad (3)$$

In situations where only partial competition was observed, the normalized concentration-response curves were fit using Eq. 1 and where $Y_{min}$ was constrained to >0.

## Intracellular fluorescence

HEK cells were transfected as described previously for the NanoBRET assay. $3 \times 10^5$ cells/mL in DMEM (Life Technologies) + 10% FBS (Life Technologies) medium were seeded in a glass bottom Fluorodish 35 mm (World Precision Instruments) and settled ON at 37 °C and 5% $CO_2$ atmosphere. After 24 h the medium was aspirated, and the cells were fixed with 4% formaldehyde (Thermo Scientific) in 1x PBS for 15 min. The solution was aspirated, and the cells were washed three times with 1 mL of 1x PBS for 5 min each. The cells were blocked with 1 mL blocking solution (1x PBS, 1% BSA/0.1% saponin, prepared freshly) for 1 h at RT. After aspirating the blocking solution, the following primary antibodies were applied as a 1:50 dilution in blocking solution and incubated standing ON at 4 °C: PRMT5 (ab109451, Abcam) and NL-PRMT5 (N7000, Promega). After repeating the wash step with 1x PBS three times, the secondary antibodies were applied in blocking solution as 1:50 dilutions for 1 h at RT as follows: Donkey Anti-Mouse IgG H&L (Alexa Fluor® 568) (ab175472, Abcam) and Goat Anti Rabbit IgG H&L (Alexa Fluor® 488) (ab150077, Abcam). The solution was aspirated, and the samples kept under 1x PBS. If applicable, for nuclei stain ProLong™ Glass Antifade Mountant with SYTOX™ Deep Red (P36993, Thermo Scientific) was applied to the cells as one drop and the samples were kept at 4 °C until measurement. Fluorescent micrographs were acquired on a Zeiss LSM 710 META confocal laser-scanning microscope using a 40x/1.3 objective.

## BRET probe sensitivity to SAM and MTA levels

To determine the linear range of SAM detection, HEK293 cells expressing NLuc-PRMT5 were lysed by addition of a 10x digitonin solution (Sigma), 500 µg/mL in Opti-MEM) and treated with a 1 mM, 4-fold, 11 pt dose-response series of SAM (Sigma) and a 1 µM 2-fold, 8 pt dose-response series of CBH-002. Lysate was incubated at RT for 30 min prior to reading BRET. Data was normalized using Eq. (2) above, with $BRET_{MIN}$ = BRET signal at lowest [CBH-002] + 0 µM SAM, and $BRET_{MAX}$ = BRET signal at 1 µM CBH-002 + 1 mM SAM. Data were fit using Eq. (3), and CBH-002 $EC_{50}$ was plotted against [SAM] to determine the linear range of detection.

To determine BRET probe binding kinetics, live HEK293 cells were treated with 30 nM CBH-002 and BRET measurements were read in 3 min intervals for 2 h at room temperature. A 20 µM, 3-fold, 11 pt dose-response series of GSK3326595 (MedChem Express) was then added to wells and the plate was read at 3 min intervals for an additional 3.5 h at room temperature.

To measure BRET probe sensitivity to both MAT2A inhibition and exogenous MTA in live cells, HEK293 cells were pre-treated with 30 nM CBH-002 and incubated at 37 °C and 5% $CO_2$ for 2 h. For MAT2A inhibition, a 20 µM, 3-fold, 11 pt titration series of Compound A (Selleckchem) was added to wells. Cells were incubated either at 37 °C (Fig. 3D) or room temperature (Fig. S3D). Data were normalized using Eq. (2), where $BRET_{MIN}$ = instrument background (no CBH-002), and $BRET_{MAX}$ = 2 h vehicle treatment, and fit curves were generated using Eq. (1). For MTA sensitivity cells were treated with a 1 mM, 11 pt, 2-fold series of MTA (Sigma) and cells were incubated at room temperature for 2 h, reading in 15 min intervals. Data were normalized using Eq. (2), where $BRET_{MIN}$ = instrument background (no CBH-002), and $BRET_{MAX}$ = 2 h + vehicle treatment. Data were fit with Eq. (1).

## Compound $EC_{50}$ titrations against MTA

HEK293 cells or wild-type HCT116 cells expressing NLuc-PRMT5 and untagged WDR77 DNA were treated with a fixed concentration of 30 nM CBH-002, as well as a matrix of inhibitor and MTA concentrations. MTA (Sigma) was prepared at $100\times$ in 100% DMSO and diluted to $10\times$ in Opti-MEM. Inhibitors were run as 11 pt, 3-fold dilution curves and MTA as a seven point, 3-fold dilution curve starting at 100 μM. Cells were incubated for 2 h at 37 °C and 5% $CO_2$ prior to BRET measurements. Data was normalized using Eq. (2) where $BRET_{MIN}$ = saturating compound + maximum [MTA], and $BRET_{MAX}$ = BRET signal under vehicle conditions. Curves were fit using Eq. (3), and $EC_{50}$ values were plotted against MTA concentration to determine the linear range of MTA detection.

## Global data analysis

For compounds in which the addition of MTA decreased affinity and maintained a constant Hill slope, the titration data was also globally analyzed using the Gaddum/Schild $EC_{50}$ shift equation in GraphPad Prism (10.5)[38]:

$$Y = Y_{min} + \frac{Y_{max} - Y_{min}}{1 + 10^{(LogEC-X)h}} \qquad (4)$$

Where $\qquad LogEC = Log(EC50 \cdot Antag),\qquad EC50 = 10^{LogEC50}\qquad$ and $Antag = 1 + (\frac{B}{10^{-pA2}})^S$

$EC_{50}$ is the compound affinity at [MTA] = 0, h is the Hill Slope, pA2 is the negative logarithm of the concentration of MTA needed to shift the curve by a factor of 2 and the Schild Slope (S) quantifies how well the effect of MTA on compound binding approximates competitive binding. Constraining the Schild slope to 1 resulted in a value for $K_b$, the affinity of MTA for PRMT5.

For compounds in which the addition of MTA caused an increased in affinity and alteration in Hill slope, the data were fit to an equation with two quadratic functions each based on the Morrison equation[39]:

$$Y = 50\left(1 - \frac{([E]_T + X + K_i) - \sqrt{([E]_T + X + K_i)^2 - 4[E]_T X}}{2[E]_T}\right)$$
$$+ 50\left(1 - \frac{([E]_T + X + K_{ii}) - \sqrt{([E]_T + X + K_{ii})^2 - 4[E]_T X}}{2[E]_T}\right) \qquad (5)$$

$E_t$ is the total enzyme concentration, while $K_i$ and $K_{ii}$ are the equilibrium constants of the compound for two non-equivalent binding sites.

## MRTX1719 kinetic washout experiment

HEK293 cells were cultured and transfected with NL-PRMT5 and WDR77 (Promega) as described previously, and incubated at 37 °C and 5% $CO_2$ overnight. On the next day, cells were resuspended in a 15 mL falcon tube and adjusted to $2 \times 10^5$ cells/mL in assay medium (Opti-MEM with 1% FBS). Final concentrations of MTA (100 μM, 0 μM), compound MRTX1719 (10 μM, 20 nM) or DMSO control were added to the falcon tubes and incubated for 2 h at 37 °C and 5% CO2. After incubation, cells were centrifuged at $300 \times g$ for 5 min, medium was aspirated and cells were washed with pre-warmed assay medium, centrifuged again and adjusted to $2 \times 10^5$ cells/mL. For the full occupancy control, MTA and compound concentrations were maintained. 34 μL of cells were added per well in a seeded into 96-well non-binding surface-treated plates (Corning), followed by addition of 36 μL of Complete 2x Substrate (prepared by combining Nano-Glo Substrate and Extracellular NanoLuc Inhibitor in Opti-MEM, according to the manufacturer's protocol (Promega)). Then, 2 μL of ETP CBH-002 at a final concentration of 30 nM were added before immediate readout at

the PheraSTAR FSX for 6 h. Readout and analysis were conducted as described earlier.

## Reporting summary

Further information on research design is available in the Nature Portfolio Reporting Summary linked to this article.

## Data availability

All data presented here are available within the paper and its Supplementary Information and Source Data files. The proteomics data generated in this study have been deposited in the PRIDE database under accession code PXD028138. Source data are provided with this paper.

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

## Acknowledgements

E.M.R., J.A.W., and K.V.M.H. are grateful for support from Myeloma UK and Bayer AG, and P.J.T. is grateful for support from the National Institutes of Health (GM149297). This project has received funding from the Engineering and Physical Sciences Research Council (EPSRC) and the Medical Research Council (MRC) [grant number EP/L016044/1] as well as the Innovative Medicines Initiative 2 Joint Undertaking (JU) under grant agreement No 875510. The JU receives support from the European Union's Horizon 2020 research and innovation program and EFPIA and Ontario Institute for Cancer Research, Royal Institution for the Advancement of Learning McGill University, Kungliga Tekniska Hoegskolan, Diamond Light Source Limited. CBH-001 bulk synthesis was performed by Enamine. KMS-11 cells were kindly provided by M. Kaiser, ICR (Institute of Cancer Research). The authors thank E. Kennedy, J. Bennett, O. Fedorov, and B. Kessler for technical support and discussions. The authors thank S. Bonham, S. Hester and R. Fischer from the TDI (Target Discovery Institute) Discovery Proteomics Facility for their assistance with MS sample processing.

## Author contributions

E.M.R., A.M., U.S., J.D.V., M.T.B., B.B.M., C.M.R., and J.W. designed, executed and analyzed BRET experiments. E.M.R. synthesized CBH-002. E.M.R., J.A.W., and J.S. conducted proteomic experiments and analyzed MS data. E.B. performed crystal structure analysis. J.H. contributed IF experiments. P.L. cloned the PRMT7 and PRMT9 constructs. P.J.T., A.W., and J.T.-J. analyzed the data. P.E.B. contributed to supervision of experiments and data discussions. E.M.R., A.M., A.W., P.J.T., M.B.R., and K.V.M.H. wrote the manuscript with contributions from all authors. M.B.R. conceived the experimental strategy for probing MTA/MTAP-dependent mechanisms and together with K.V.M.H. jointly supervised the study. K.V.M.H. designed the compounds, acquired funding, and conceived the overall study. All authors have given approval to the final version of the manuscript.

## Competing interests

A.M., J.D.V., M.T.B., B.B.M., J.W., and M.B.R. are employees of Promega, and Promega owns patents related to NanoLuc and Nano-BRET technologies. The remaining authors declare no competing interests.
