## [Transparent Peer Review file · Nature Communications]

A BRET Biosensor for Measuring Uncompetitive Engagement of PRMT5 Complexes in Cells

Corresponding Author: Professor Kilian Huber

Version 1:

Reviewer comments:

Reviewer #1

(Remarks to the Author)

The authors present the development of a BRET assay capable of measuring target engagement of compounds that bind to PRMT5. Due to the selection of the probe compound, this assay is responsive to the intracellular concentrations of the PRMT5 co-substrate, SAM, or a competing metabolite, MTA, which will enable this assay to contribute to the development of PRMT5 inhibitors that are potentiated by MTAP deletion. This report may also inspire others to develop BRET assays that monitor target engagement in a metabolite responsive manner.

The interaction between GSK'595 and SAM as described in Figure S6 seems crucial to CBH-002's responsiveness to the concentration of SAM/MTA. The manuscript will be easier to understand if this information is incorporated into the results section associated with Figure 3 rather than waiting until the discussion to bring in this information.

In the discussion, please clarify the comparison to BRET probes of kinases. Does "direct measurement of uncompetitive target engagement" refer to MTA-uncompetitive target engagement? This statement seems more accurate: An uncompetitively binding probe can be successfully developed into a BRET probe for uncompetitive target engagement.

Please incorporate a comment about relative WDR77 expression in the discussion section comparing MTA dependence of MRTX1719 affinity in HEK293 vs. HCT116 unless WDR77 overexpression was also performed in HEK293.

To help increase the relevance of this work beyond its current target protein, it would be helpful to add more commentary to the discussion section about the choice to elaborate GSK'595 into a BRET probe rather than from other ligands profiled in this manuscript, such as MRTX1719. Would MRTX1719 also be expected generate a successful SAM/MTA biosensor and/or MTA-uncompetitive PRMT5 probe? Many of the conclusions around the assay serving as a biosensor depend on the properties of the chosen probe compound, so please bring the description of the probe properties and the discussion of the novelty of the current work into the same section of the discussion for clarity.

Minor points:

In the methods section, please include more details about the mass spectrometer used to acquire the thermal profiling data. Please update the PRIDE description to include a brief description of the thermal profiling assay and also please deposit the MaxQuant output associated with the chemical proteomics data. The section referring to STRING analysis can likely be removed as protein interaction networks are not described.

Figure S6 is very crucial to the understanding of the properties of CBH-002. Please add the references for Fig. S6 to the main text.

Reviewer #2

(Remarks to the Author)

Summary:

In the manuscript by Rothweiler and colleagues they describe a novel BRET-based tool to measure both PRMT5 inhibitor

and co-factor metabolite binding in cells expressing their PRMT5 reporter. The results provided are of high interest given the numerous PRMT5 inhibitors being developed and large number of cancer patients who could potentially benefit from these targeted therapies. Additionally, these results provide a unique and interesting example of a BRET tool that can be used simultaneously to quantify direct and indirect mediators of target engagement. The manuscript is lacking in a few key areas in its current form, however, to be suitable for publication in Nature Communications and revisions for the following points would significantly strengthen it.

Major points:

1. The manuscript would benefit from additional evidence on what MTAP deletion does to endogenous levels of MTA and SAM metabolites in HCT116 WT vs MTAP^{-/-} cells under the BRET assay conditions to provide better context to the changes of potency seen with MRTX1719 in figure 5F. The MTA addition experiments in figures 4 and S4 are of such high concentration and broad range that it would be especially useful to know how the actual endogenous levels compare. As the authors also state in the discussion, the ability of this BRET assay to sense both inhibitor and metabolite binding could make it a valuable tool, therefore directly showing what the SAM and MTA levels are under the assay conditions is critical.
2. In figure 3D the authors show that MAT2A inhibition by compound A alone can fully reduce the normalized BRET signal to zero. In fact, MAT2A inhibition seems to be of similar potency as some of the direct PRMT5 inhibitors shown in figures 2D-F. Is that correct? It is hard to believe an indirect inhibitor to PRMT5 could have such a similar effect!? Or are these comparisons misleading due to the normalization that was done?
3. The authors speculate that intracellular SAM/MTA pool may vary between MTAP-deleted cancer cells and result in different potencies for MRTX1719 and TNG908. Can't the authors test this directly here with their PRMT5 BRET system to be less speculative and instead provide evidence for both the baseline metabolite level effects on BRET signal and then how potent these inhibitors are in that context?

Minor points:

1. Abstract refers to approximately 10% of tumors harboring MTAP homozygous deletions, the introduction states it is 10-15%.
2. The authors demonstrate that metabolites SAM or MTA affect uncompetitive PRMT5 inhibitor binding to PRMT5 in their assay. Is the assay also capable of looking at kinetics to determine how on an off rates of inhibitors are more specifically effected by metabolite levels?

Reviewer #3

(Remarks to the Author)

General remarks:

Rothweiler et al. present the development and validation of a new optical tool, CBH-002, enabling BRET-based binding measurements with methyltransferase PRMT5. Furthermore, the authors show how this new biosensor can be used to understand the binding mechanism of different PRMT5 ligands engaging different sites of the target enzyme. Overall, the data is very convincing. The presented datasets provide deep insights into the pharmacological and mechanistic action of different PRMT5 ligands, into the effect of distinct cellular levels of methionine metabolites MTA and SAM and into the effect of disease-associated mutations of enzymes involved in methionine metabolism in different cell lines. Certainly, this new tool will be extremely useful and the new tool should be extremely useful for future screening and pharmacological characterization of future PRMT5 modulators. My only major criticism refers to the fact that the authors do not provide any time-course data throughout the manuscript. One of the major advantages of BRET compared to many other assay formats, is that the process of interest can be monitored in real-time. This advantage should be exploited since additional mechanistic insights can be obtained from kinetic analyses of BRET responses in living cells.

Major comments:

- BRET time course data, representing the real-time binding kinetics of CBH-002 to PRMT5, should be provided for the key experiments of this study. Along these lines, it would be important to see the real-time displacement time course of the concentration response curves presented in Figure 3D. Upon pre-incubation to a BRET equilibrium with 5 nM CBH-002, MTA should show a much faster displacement effect compared to compound A, which does not directly bind to PRMT5 but blocks the enzymatic production of SAM.

Minor comments/notes:

- It should be shown whether the new BRET probe can be used to determine the binding mode of the different PRMT5 inhibitor classes. Therefore a Schild analysis for competitive vs. uncompetitive cooperativity with SAM, MTA or peptide substrates should be performed. Since the BRET probe is based on a peptide-substrate competitive inhibitor of PRMT5, it should yield to a linear relationship with a hill slope = 1 for another peptide substrate (as shown in figure 1C for the peptide-substrate competitive inhibitor GSK3326595). Displacement experiments with SAM, MTA or other co-substrate competitive inhibitors should, however, result in a non-linear relationship in a Schild plot analysis due to the allosteric cooperativity.
- No abbreviations without explanation (MTA; MTAP) in the abstract without explanation.
- The introduction is too long and should lead more quickly and precisely to the research question and the aim of the study presented
- Line 93: remove "(refs)".
- Figure S2 and lines 144 – 146: the second phase of the concentration response curve could also indicate a non-specific BRET increase due to, e.g., radiative excitation of the BRET probe by nanoluc at high CBH-002 concentrations. This is supported by the increase in BRET at a similar concentration range for Nluc-fused PRMT7 and PRMT9 and should be further tested by adding CBH-002 to a non-PRMT, cytosolic and Nluc-fused protein.

Version 2:

Reviewer comments:

Reviewer #1

(Remarks to the Author)

The authors have improved the clarity and impact of their manuscript through revision and I support publication of this version.

Reviewer #2

(Remarks to the Author)

The authors have addressed all of my previous concerns and I congratulate them on a wonderful and impactful story!

Reviewer #3

(Remarks to the Author)

All my concerns have been addressed. The new data showing the BRET time courses and the Schild analysis improves the overall manuscript quality. I'd like to recommend the manuscript for publication in Nature Communications.

Point-by-point response

1.1: *The interaction between GSK'595 and SAM as described in Figure S6 seems crucial to CBH-002's responsiveness to the concentration of SAM/MTA. The manuscript will be easier to understand if this information is incorporated into the results section associated with Figure 3 rather than waiting until the discussion to bring in this information.*

Response: We fully agree with the comments from reviewer 1 regarding the importance of showing the interaction between GSK3326595 and SAM as described in the re-evaluation of co-crystal structures from previous publications. We have moved the supplementary discussion to the results section (lines 230-241) and moved the figure panel forward to Figure S4.

1.1b: *In the discussion, please clarify the comparison to BRET probes of kinases. Does “direct measurement of uncompetitive target engagement” refer to MTA-uncompetitive target engagement? This statement seems more accurate: An uncompetitively binding probe can be successfully developed into a BRET probe for uncompetitive target engagement.*

Response: This passage can be modified for better clarity on the specific MoA in question. The purpose of the statement is to contrast the new capability to observe uncompetitive engagement with BRET probes, not competitive engagement (as has been applied to kinases). We are not aware of earlier studies of BRET probes for uncompetitive target engagement analysis in cells. Nonetheless, we have altered the sentence by specifying “MTA-uncompetitive engagement” to better reflect the MoA in question (line 485).

1.2: *Please incorporate a comment about relative WDR77 expression in the discussion section comparing MTA dependence of MRTX1719 affinity in HEK293 vs. HCT116 unless WDR77 overexpression was also performed in HEK293.*

Response: Following the comments from reviewer 1 and 2 regarding WDR77 expression in HEK and HCT116 to elucidate MTA dependence of MRTX1719 affinity in cells, we performed additional experiments with WDR77 co-expression in HEK and have updated the text (lines 121-128) and Figure 2 accordingly. Using NL-PRMT5 alone led to a

biphasic dose response, whereas the addition of untagged WDR77 potentiated the signal by orders of magnitude supporting the formation of a PRMT5-WDR77 complex. Under co-expression, a single binding isotherm was observed, suggesting the complex may be favoured under WDR77 saturating levels.

1.3: To help increase the relevance of this work beyond its current target protein, it would be helpful to add more commentary to the discussion section about the choice to elaborate GSK'595 into a BRET probe rather than from other ligands profiled in this manuscript, such as MRTX1719. Would MRTX1719 also be expected generate a successful SAM/MTA biosensor and/or MTA-uncompetitive PRMT5 probe? Many of the conclusions around the assay serving as a biosensor depend on the properties of the chosen probe compound, so please bring the description of the probe properties and the discussion of the novelty of the current work into the same section of the discussion for clarity.

Response: To address reviewer 1's comment we elaborate on the choice of GSK3326595 as a probe compound in the results section (lines 425-428) and hypothesize about MRTX1719 to generate a similarly successful SAM/MTA biosensor in the discussion (lines 454-459).

1.4: In the methods section, please include more details about the mass spectrometer used to acquire the thermal profiling data. Please update the PRIDE description to include a brief description of the thermal profiling assay and also please deposit the MaxQuant output associated with the chemical proteomics data. The section referring to STRING analysis can likely be removed as protein interaction networks are not described. Figure S6 is very crucial to the understanding of the properties of CBH-002. Please add the references for Fig. S6 to the main text.

Response: We thank reviewer 1 for bringing these points to our attention and have updated the details about the mass spectrometer for the thermal profiling data in the methods section (lines 651-652) and have updated the PRIDE description accordingly. We have uploaded the MaxQuant output proteinGroups.txt file to the PRIDE submission. We have removed the STRING analysis description section. Figure S6 has

now become Figure S4 and the references (38, 39) have been moved to the main text (lines 230-241, cf. comment 1.1).

2.1: *The manuscript would benefit from additional evidence on what MTAP deletion does to endogenous levels of MTA and SAM metabolites in HCT116 WT vs MTAP^{-/-} cells under the BRET assay conditions to provide better context to the changes of potency seen with MRTX1719 in figure 5F. The MTA addition experiments in figures 4 and S4 are of such high concentration and broad range that it would be especially useful to know how the actual endogenous levels compare. As the authors also state in the discussion, the ability of this BRET assay to sense both inhibitor and metabolite binding could make it a valuable tool, therefore directly showing what the SAM and MTA levels are under the assay conditions is critical.*

Response: This is a valuable comment and relates to question 2.3 below. We have therefore combined our response to both queries below.

2.2: *. In figure 3D the authors show that MAT2A inhibition by compound A alone can fully reduce the normalized BRET signal to zero. In fact, MAT2A inhibition seems to be of similar potency as some of the direct PRMT5 inhibitors shown in figures 2D-F. Is that correct? It is hard to believe an indirect inhibitor to PRMT5 could have such a similar effect!? Or are these comparisons misleading due to the normalization that was done?*

Response: The reviewer raises a good point. To address this comment and also the comment from Reviewer 3.1, we have repeated these experiments under more physiological conditions to determine the real-time impacts of MTA binding to PRMT5, and also real-time impacts of upstream inhibition of SAM levels on the biosensor readout. As expected, relatively rapid kinetic effects of MTA treatment were observed on the biosensor, which plateaued within 2 hours at room temperature. In contrast, the effect of MAT2A inhibition was considerably slower, and required 37 °C, to manifest its effect on the PRMT5 biosensor. We have also provided alternate normalization of the data shown, supporting fractional effects at a subpopulation of PRMT5. We have included these new results in the figures 3D, 3E, S3C, and S3D. Please refer to lines 201-234 for text revisions.

2.3 *The authors speculate that intracellular SAM/MTA pool may vary between MTAP-deleted cancer cells and result in different potencies for MRTX1719 and TNG908. Can't the authors*

test this directly here with their PRMT5 BRET system to be less speculative and instead provide evidence for both the baseline metabolite level effects on BRET signal and then how potent these inhibitors are in that context?

Response: This is an excellent question. Motivated by this and Reviewer comment 2.1, we have used the BRET biosensor to explore the intracellular concentrations of SAM and MTA in all three cell lines used in our study (HEK293, HCT116-WT, and HCT116 MTAP-/-). First, since CBH-002 binding is expected to be dependent on SAM, we adapted the BRET biosensor to measure SAM concentrations in cell lysates. A linear response to SAM was detected over a broad range of concentrations. Using the potency of the CBH-002 BRET probe, we were able to estimate that SAM concentrations in these cell lines were fairly consistent, regardless of MTAP status. However, MTA levels in MTAP null vs wildtype are likely far more dynamic, and CBH-002 did not demonstrate the same level of sensitivity to exogenous [MTA]. To better estimate MTA levels across our cell lines in the study, we instead titrated exogenous MTA and measured its impact on MRTX1719 potency. To determine the linear range of MTA sensitivity to MRTX1719 potency, we performed MTA titrations in HCT116 wildtype cells. Gratifyingly, MTA sensitivity was established over multiple orders of magnitude. Using MRTX1719 as a calibrator, we were therefore able to estimate the relative impact of MTAP ablation on MTA levels, which could serve as powerful assay for other cell lines in the future. We thank the reviewer for this recommendation and have included new data. Please see text revisions in lines 190-200, 353-381, 459-483, Figures 5, S3, S5, S6 and Table S5, S6.

2.4: Abstract refers to approximately 10% of tumors harboring MTAP homozygous deletions, the introduction states it is 10-15%.

Response: We thank reviewer 2 for highlighting this and have corrected this accordingly in line 22 in the abstract, including the corresponding reference in the introduction.

2.5: The authors demonstrate that metabolites SAM or MTA affect uncompetitive PRMT5 inhibitor binding to PRMT5 in their assay. Is the assay also capable of looking at kinetics to determine how on an off rates of inhibitors are more specifically effected by metabolite levels?

Response: This is a valuable point, and warranted further experimentation. Indeed, through washout experiments, BRET probes can be used to determine inhibitor off-rates when the probe itself has sufficiently fast intrinsic binding kinetics. Motivated by this query, we explored the association kinetics of CBH-002 in live cells and its compatibility with residence time analysis. Although CBH-002 demonstrated slow association kinetics under the optimized conditions of the assay (30 nM BRET probe), we were able to increase the concentration and accelerate the apparent association kinetics, supporting qualitative assessments of drug residence time. To explore the impact of [MTA] on MRTX1719 off-rate, we tested conditions of low [MTA] (requiring high concentrations of MRTX1719) and high [MTA] (requiring low concentration of MRTX1719). As shown in Figure S5I, saturation of the PRMT5 biosensor with MTA resulted in protracted binding, with only fractional dissociation occurring over 6 hours of analysis. These data provide the first evidence we are aware of that the MTA-bound PRMT5 complex has more durable engagement of MTA-cooperative inhibitors in cells. This feature may offer improved therapeutic index in MTAP deleted cancers. We thank the reviewer for motivating us to do these experiments. Please see lines 331-339 for text additions.

3.1. BRET time course data, representing the real-time binding kinetics of CBH-002 to PRMT5, should be provided for the key experiments of this study. Along these lines, It would be important to see the real-time displacement time course of the concentration response curves presented in Figure 3D. Upon pre-incubation to a BRET equilibrium with 5 nM CBH-002, MTA should show a much faster displacement effect compared to compound A, which does not directly bind to PRMT5 but blocks the enzymatic production of SAM.

Response: We are grateful for this recommendation and have performed additional experiments to address this comment, and also the comments from Reviewer 2.2 (above). Real time studies support the hypothesis above; MTA effects were observed at short treatment times, whereas MAT2A inhibition effects required longer treatment times at physiological temperatures. We have included new data in the supplementary and main body (Figure 3E, Figure S3D) in support of these findings. Please see text revisions in lines 201-229.

3.2 *It should be shown whether the new BRET probe can be used to determine the binding mode of the different PRMT5 inhibitor classes. Therefore a Schild analysis for competitive vs. uncompetitive cooperativity with SAM, MTA or peptide substrates should be performed. Since the BRET probe is based on a peptide-substrate competitive inhibitor of PRMT5, it should yield to a linear relationship with a hill slope = 1 for another peptide substrate (as shown in figure 1C for the peptide-substrate competitive inhibitor GSK3326595). Displacement experiments with SAM, MTA or other co-substrate competitive inhibitors should, however, result in a non-linear relationship in a Schild plot analysis due to the allosteric cooperativity.*

Response: We thank the reviewer for suggesting a more rigorous analysis. In lines 298-316, we have now added a global analysis of the data using the Gaddum/Schild EC50 equation using GraphPad Prism (Fig. S5G, Table S3). This analysis was performed for the cosubstrate competitive (LLY-283 and PF-06939999) and SAM-uncompetitive inhibitors (GSK3326595 and EPZ015666), where addition of MTA reduces compound affinity without changing the Hill slope. In each case the Schild slope is close to 1 consistent with competitive binding. Subsequent constraint of the slope to 1 provides a value for the affinity of MTA for PRMT5 (K_b). For the two MTA uncompetitive inhibitors where MTA increases the apparent affinity we have now globally fit the data to an equation that is comprised of two quadratic equations to account for the change in Hill slope during the titration and based on the hypothesis that at high [MTA] the titration is approaching tight binding conditions.

3.3: *No abbreviations without explanation (MTA; MTAP) in the abstract without explanation.*

Response: We thank reviewer 3 for noting the abbreviations without explanation in the abstract and have introduced those accordingly.

3.4. *The introduction is too long and should lead more quickly and precisely to the research question and the aim of the study presented*

Response: We have amended the introduction as suggested and shortened and re-structured the text accordingly.

3.5: Line 93: remove “(refs)”

Response: We have removed “(refs)” from line 93.

3.6: *Figure S2 and lines 144 – 146: the second phase of the concentration response curve could also indicate a non-specific BRET increase due to, e.g., radiative excitation of the BRET probe by nanoluc at high CBH-002 concentrations. This is supported by the increase in BRET at a similar concentration range for Nluc-fused PRMT7 and PRMT9 and should be further tested by adding CBH-002 to a non-PRMT, cytosolic and Nluc-fused protein*

Response: In response to Reviewer 3’s comment, we have included additional cytosolic and nuclear controls, RIPK2-NL and NL-BRD9, respectively. We observe non-specific BRET increases at high BRET probe concentrations and show these results in Supplementary Figure S2C-F (lines 128-130). However, we would like to highlight that within the chosen assay window at 30 nM BRET probe concentration no unspecific BRET is detected.